# Unlocking Multimodal Mathematical Reasoning via Process Reward Model

**Ruilin Luo**[12*]    **Zhuofan Zheng**[2*]    **Lei Wang**[3]    **Yifan Wang**[1]
**Xinzhe Ni**[1]    **Zicheng Lin**[1]    **Songtao Jiang**[4]    **Yiyao Yu**[1]
**Chufan Shi**[1]    **Ruihang Chu**[1†]    **Jin Zeng**[2†]    **Yujiu Yang**[1]

[1]Tsinghua University    [2]ByteDance
[3]Ping An Technology (Shenzhen) Co., Ltd.    [4]Zhejiang University

## Abstract

Process Reward Models (PRMs) have shown promise in enhancing the mathematical reasoning capabilities of Large Language Models (LLMs) through Test-Time Scaling (TTS). However, their integration into multimodal reasoning remains largely unexplored. In this work, we take the first step toward unlocking the potential of PRMs in multimodal mathematical reasoning. We identify three key challenges: (i) the scarcity of high-quality reasoning data constrains the capabilities of foundation Multimodal Large Language Models (MLLMs), which imposes further limitations on the upper bounds of TTS and reinforcement learning (RL); (ii) a lack of automated methods for process labeling within multimodal contexts persists; (iii) the employment of process rewards in unimodal RL faces issues like reward hacking, which may extend to multimodal scenarios. To address these issues, we introduce **URSA**, a three-stage **U**nfolding multimodal p**R**ocess-**S**upervision **A**ided training framework. We first construct MMathCoT-1M, a high-quality large-scale multimodal Chain-of-Thought (CoT) reasoning dataset, to build a *stronger math reasoning foundation MLLM*, URSA-8B. Subsequently, we go through an automatic process to synthesize process supervision data, which emphasizes both logical correctness and perceptual consistency. We introduce DualMath-1.1M to facilitate the training of URSA-8B-RM. Finally, we propose **P**rocess-**S**upervised **G**roup-**R**elative-**P**olicy-**O**ptimization (**PS-GRPO**), pioneering a *multimodal PRM-aided online RL method* that outperforms vanilla GRPO. With PS-GRPO application, URSA-8B-PS-GRPO outperforms Gemma3-12B and GPT-4o by 8.4% and 2.7% on average across 6 benchmarks. Code, data and checkpoint can be found at https://github.com/URSA-MATH.

## 1 Introduction

Following the substantial progress of Large Language Models (LLMs) in math reasoning [1–8], the math reasoning capabilities of Multimodal Large Language Models (MLLMs) have increasingly garnered attention [9–13]. Previous work has typically focused on aspects such as math reasoning data curation [14–18], training math-intensive vision encoders [19, 20], enhancing vision-language alignment [11, 21], or the application of post-training techniques [22–24, 13]. Given the success of Process Reward Models (PRMs) in improving LLM reasoning through methods like Test-Time Scaling (TTS) [25, 26] and Reinforcement Fine-Tuning (ReFT) [27, 28], the application of PRMs to multimodal reasoning remains unexplored.

---

* Equal contribution. Work done during Ruilin's internship at ByteDance. † Corresponding author.
ruihangchu@gmail.com, zengjin@bytedance.com

39th Conference on Neural Information Processing Systems (NeurIPS 2025).

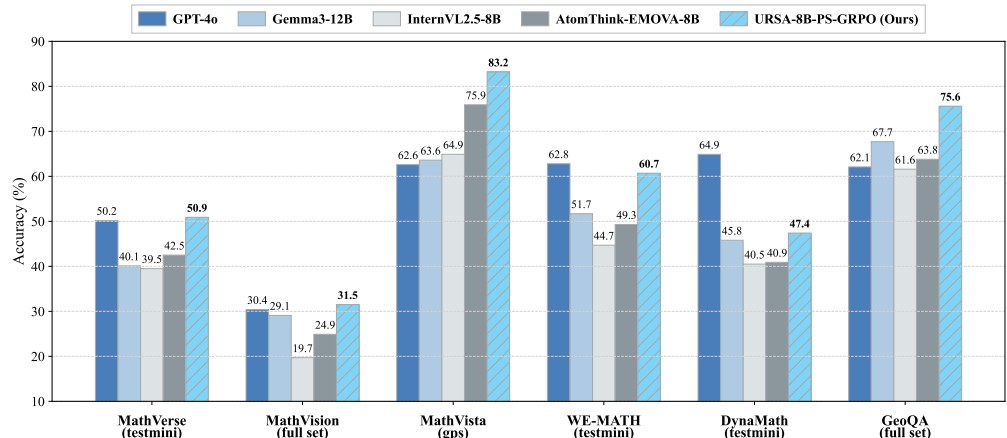

Figure 1: Performance comparison with leading open-source MLLMs and GPT-4o.

In this work, we take the first step toward integrating PRMs into multimodal math reasoning. We identify three key challenges: (i) Since both TTS and RL are heavily influenced by the strength of foundation models [29, 25], the limited availability of large-scale, high-quality reasoning data constrains the upper bounds of current MLLMs and weakens the effectiveness of PRM integration; (ii) There hasn't yet been adequate automated process labeling techniques merged within multimodal contexts, where both logical validity and perceptual consistency should be emphasized [30–32]. (iii) While PRMs can be effectively used in TTS, applying them directly in online RL introduces risks such as reward hacking and length bias in rewarding [33, 34].

To address these challenges, we propose the **URSA** framework, a three-stage **U**nfolding multimodal p**R**ocess-**S**upervision **A**ided training pipeline that supports both the construction and application of multimodal PRMs. In Stage I, we curate **MMathCoT-1M**, a large-scale, high-quality multimodal Chain-of-Thought dataset synthesized from 1.43 million open-source examples, which enhances the foundation model's reasoning capabilities through targeted instruction tuning. In Stage II, we construct **DualMath-1.1M** via a dual-view process supervised data synthesis strategy which combines a binary error locating engine and a misinterpretation insertion engine. It provides complementary signals for logical validity and visual grounding, and is used to train a process reward model. In Stage III, we analyze the limitations of scalar process reward modeling in online RL and propose **Process Supervision-GRPO (PS-GRPO)**, which mitigates reward hacking and PRM's length bias in rewarding by implicitly penalizing process-level inconsistencies during policy optimization.

Results on 6 multimodal reasoning benchmarks show that our PRM improves Best-of-N verification, surpassing self-consistency and outcome-based baselines. When used in PS-GRPO, the resulting model achieves state-of-the-art performance among open-source MLLMs of similar size. Our contributions are as follows:

- We release two large-scale open-source datasets, MMathCoT-1M and DualMath-1.1M, to address the scarcity of high-quality multimodal CoT reasoning and process supervision data.
- We propose PS-GRPO, an online reinforcement learning algorithm that incorporates multimodal PRMs by comparing the relative quality of rollouts, rather than relying on scalar reward modeling. It effectively mitigates PRM's reward hacking and length bias in rewarding.
- Experimental results show that our reward model improves both test-time verification and online training. With PS-GRPO application (Figure 1), URSA-8B-PS-GRPO outperforms Gemma3-12B and GPT-4o by 8.4% and 2.7% on average across 6 benchmarks.

## 2 Stage I: Math-Intensive Alignment and Instruction Tuning

### 2.1 Collection of Vision-Language Alignment Data

We employ a LLaVA-like architecture and first collect vision-language alignment data directly from existing open-source datasets [35–38]. As demonstrated in Figure 2, we collect URSA-Alignment-860K from Multimath [23], MAVIS [19] and Geo170K [18]. We then filter out samples with overly

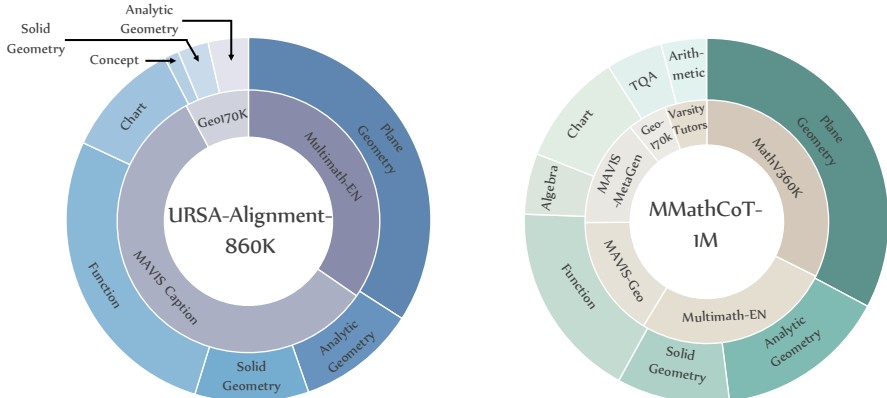

Figure 2: Statistics of URSA-Alignment-860K and MMathCoT-1M.

verbose captions, to form an 860K math-intensive alignment dataset. Following the engineering practices of previous work, we only train the MLP projector in the alignment step.

## 2.2 CoT Reasoning Data Synthesis

For a powerful foundation building, we collect 1.43M samples from existing math reasoning datasets to support the construction of large-scale CoT reasoning data. As shown in Figure 2, data is sourced from MathV360K [15], Multimath [23], MAVIS [19], Geo170K [18] and VarsityTutors [11]. Based on the type of solution, we categorize the data into *answer-only*, *analysis-formatted*, and *CoT-formatted*. We adopt different synthesis strategies for them to curate high-quality CoT reasoning trajectories. We utilize Gemini-1.5-Flash-002 (refer to $\mathcal{G}$ below) as a cost-effective tool for data curation, avoiding expensive large-scale manual annotation.

**CoT Expansion.** For *answer-only* data $\mathcal{D}_1 = \{(x_i, y_i)\}_{i=1}^{N_1}$, such as MathV360K [15], each sample contains a question $x_i$ and a ground-truth answer $y_i$. This type of data is heavily used in previous works for fast thinking reasoning mode [15, 11, 16]. However, answer-only training restricts the model from fully capturing the problem-solving process. It may lead to memory-based reasoning, hindering the model's ability to directly provide answers to more complex reasoning problems [39]. We expand certain scale CoT reasoning trajectories for this category of data. Given a expansion prompt $\mathcal{P}_\mathcal{C}$, we provide $x_i$ and $y_i$, then prompt $\mathcal{G}$ to output the reasoning trajectory leading to the answer $y_i$, yielding the expanded solutions $\mathcal{S}_{Ao} = \mathcal{G}(\mathcal{P}_\mathcal{C}; \{x_i, y_i\}_{i=1}^{N_1})$.

**Rewriting.** This strategy is designed for *analysis-formatted* samples, denoted as $\mathcal{D}_2 = \{(x_i, y_i, a_i)\}_{i=1}^{N_2}$. This includes datasets like MAVIS-Geo, MAVIS-MetaGen [19], VarsityTutors [11], and Geo170K-QA [40]. Each sample contains a question $x_i$, an answer $y_i$, and textual analysis $a_i$. While this type of data provides walkthroughs, it often suffers from two issues: (i) It lacks strict step-by-step logic, exhibiting jumps in language or reasoning. (ii) A significant portion of the answers are relatively brief and cannot provide rich rationale. Given a rewriting prompt $\mathcal{P}_\mathcal{R}$, we utilize $\mathcal{G}$ to transcribe these solutions, thereby enhancing their step-by-step reasoning trajectories and linguistic diversity, resulting in the rewritten set $\mathcal{S}_{An} = \mathcal{G}(\mathcal{P}_\mathcal{R}; \{x_i, y_i, a_i\}_{i=1}^{N_2})$.

**Format Unification.** This strategy is used for *CoT-formatted* data, primarily sourced from Multimath-EN-300K [23], which is collected from K-12 textbooks and contains mathematical language and symbolic-style reasoning solutions. This portion of the data, $\mathcal{D}_3 = \{(x_i, y_i, c_i)\}_{i=1}^{N_3}$, consists of a question $x_i$, an answer $y_i$, and a solution $c_i$. We unify the format through natural language stylization using a prompt $\mathcal{P}_\mathcal{F}$ with $\mathcal{G}$, producing the unified set $\mathcal{S}_C = \mathcal{G}(\mathcal{P}_\mathcal{F}; \{x_i, y_i, c_i\}_{i=1}^{N_3})$.

**MMathCoT-1M.** Finally, we filter out instances where: (i) Correctness is violated: the generated content altered the original answer, or (ii) Consistency is problematic: the solution includes text that questions the original answer or makes new assumptions to force the given answer. This process yields MMathCoT-1M. The complete prompt designs can be found in Appendix H.

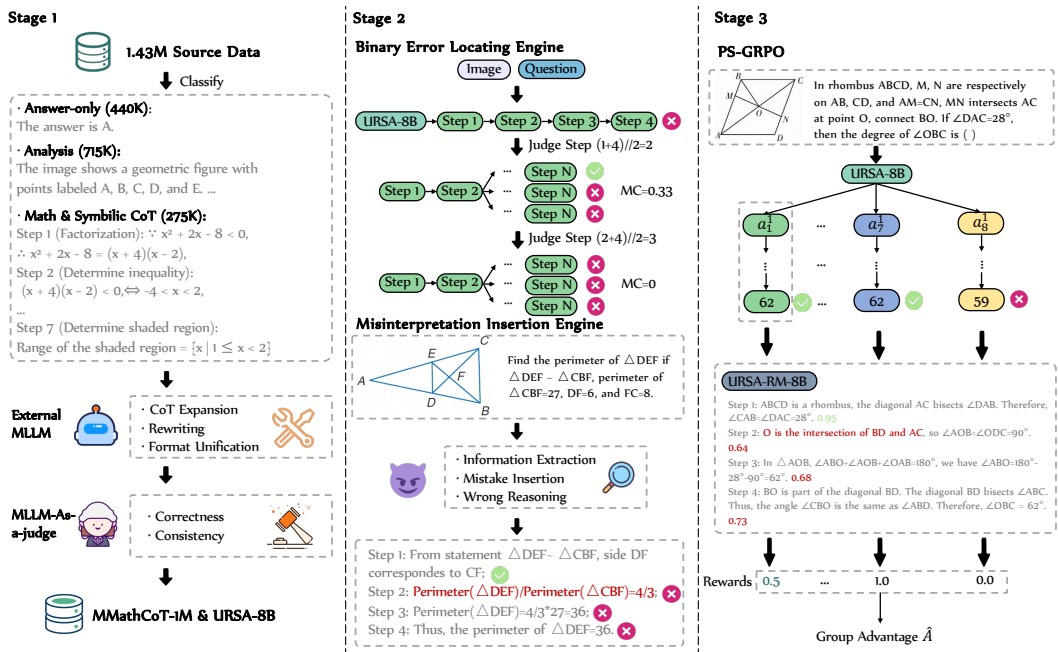

Figure 3: Pipeline of URSA. Stage 1 depicts the workflow of data curation as described in Section 2. Stage 2 illustrates how binary error locating and misinterpretation insertion facilitate the automation of process supervision data. Stage 3 demonstrates how our PS-GRPO operates by imposing penalties on rollouts that are questioned by the PRM.

We perform full-parameter instruction fine-tuning with MMathCoT-1M to train URSA-8B, based on the aligned model. The SFT dataset $\mathcal{D}_{SFT}$ is formed by the union of the curated solutions, i.e., $\mathcal{D}_{SFT} = \{(x_i, y_i) \mid (x_i, y_i) \in \mathcal{S}_{Ao} \cup \mathcal{S}_{An} \cup \mathcal{S}_C\}$. Training objective is demonstrated in Equation 1.

$$\mathcal{L}_{SFT} = -\mathbb{E}_{(x,y) \sim \mathcal{D}_{SFT}} \sum_{t=1}^{T} \log \mathcal{M}(y_t | x, y_{<t}) \tag{1}$$

In this phase, we construct a stronger reasoning foundation model, URSA-8B, with the expectation of achieving a higher bound at inference time and to process supervision data of greater diversity.

## 3 Stage II: Dual-View Process Supervised Data Synthesis

### 3.1 Binary Error Locating Engine

Following suggestions by previous work [41–43], we train a PRM for first error step identification. We collect ~553K incorrect solutions from URSA-8B's zero-shot inference on MMathCoT-1M. Erroneous steps in these solutions are labeled using Monte Carlo Tree Search (MCTS). For MCTS, an operation $\mathcal{F}(\{s_1, \ldots, s_i\}, N)$ generates $N$ rollouts from a reasoning prefix $\{s_1, \ldots, s_i\}$. The single step's Monte Carlo estimation value, $mc_i$, is the fraction of these rollouts leading to a correct answer:

$$mc_i = \frac{|\text{Correct rollouts from } \mathcal{F}(\{s_1, s_2, \ldots, s_i\}, N)|}{|\text{Total rollouts from } \mathcal{F}(\{s_1, s_2, \ldots, s_i\}, N)|} \tag{2}$$

A step $s_i$ is deemed "potentially correct" if $mc_i > 0$ [43, 42]. We optimize the identification of first error step using Binary Error Locating Engine (BEL): if the middle step has positive $mc$ (i.e. $mc_{mid} > 0$), the error is in the latter half; otherwise, in the first (see Algorithm 1). To mitigate step-level label bias and include positive examples, we add ~180K correct solutions (1/3 the number of incorrect ones), with all steps easily marked "True". This yields $\mathcal{S}_{BEL}$, a 773K process annotation dataset based on correctness potential.

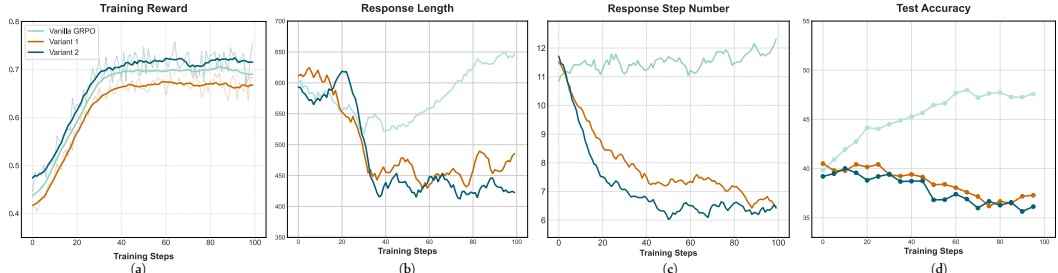

Figure 4: Figure (a)-(d) respectively illustrate training rewards, response length, response step number and test set accuracy of vanilla GRPO and two variants proposed in Section 4. Test set is randomly selected 500 examples from MMathCoT-1M for an in-domain evaluation.

## 3.2 Misinterpretation Insertion Engine

Apart from logical errors, the perception inconsistency between images and text in reasoning steps is a unique problem in multimodal scenarios [30, 44, 45]. We propose a Misinterpretation Insertion Engine (MIE) to artificially insert hallucinatory information, automatically constructing process supervision data with incorrect reasoning paths starting from the insertion point. Specifically, MIE includes three steps. First, we prompt $\mathcal{G}$ to perform a captioning task, extracting mathematical paradigm information from the image as much as possible. Second, the model $\mathcal{G}$ is required to focus on potentially confusable conditions within the existing correct solution and modify them using adjacent or similar conditions. Finally, the model $\mathcal{G}$ is prompted to continue reasoning based on the step with the inserted error. We leverage strong instruction-following capability of $\mathcal{G}$, instructing it to automatically assign negative labels to every subsequent step following the erroneous insertion. We generate ∼302K samples $\mathcal{S}_{MIE}$ using this strategy. Cases from MIE can be found in the Appendix I.2.

## 3.3 PRM Training

As shown in Equation 3, we merge two types of data, proposing a ∼1.1M process supervision data called DualMath-1.1M. During training, we append a special token after each step to indicate its predicted correctness. We model the PRM training as a binary classification task for the correctness of each step, as shown in Equation 4, here $\pi_p$ is the trained PRM based on URSA-8B. $e_j$ and $y_j$ represent single step and corresponding label ($y_j \in \{0, 1\}$).

$$\mathcal{D}_{PRM} = \{(e, y_e) \sim \mathcal{S}_{BEL} \cup \mathcal{S}_{MIE}\} \tag{3}$$

$$\mathcal{L}_{PRM} = -\mathbb{E}_{(e,y)\sim\mathcal{D}_{PRM}} \sum_{j=1}^{|e|} \left[ y_j \log \pi_p(e_j) + (1 - y_j) \log(1 - \pi_p(e_j)) \right] \tag{4}$$

Thus, Stage II delivers URSA-8B-RM, a strong PRM trained on DualMath-1.1M—the **first** large-scale, automatically labeled dataset for multimodal reasoning process supervision. While BoN evaluation demonstrates PRM's value in TTS, a critical question emerges: how can its guidance be directly integrated into MLLM post-training? This remains largely uncharted. Stage III draws a lesson about why previous scalar process reward modeling tends to fail, and then we achieve effective progress through process-as-outcome reward modeling.

## 4    Stage III: Integrating multimodal PRM into RL

Inspired by successes like DeepSeek-R1 [46], several recent studies have tried to adapt outcome reward-based GRPO for multimodal reasoning, demonstrating notable progress [47–50]. Outcome reward-based GRPO computes the $i$-th response's advantage through normalizing in-group rewards. However, outcome reward-based GRPO ignores the quality of reasoning processes [41, 51, 52].

Following most standard response-level and step-level reward modeling in RL [43, 28, 46, 13, 53], we examine two simple variants of GRPO with integrated scalar process rewards to reveal the failure patterns during the training process [54]. *Variant 1*: For $i$-th rollout, the reward is the sum of the outcome reward and the average process reward, i.e. $r^i = r_o^i + \bar{r}_s^i$. *Variant 2*: Despite the outcome

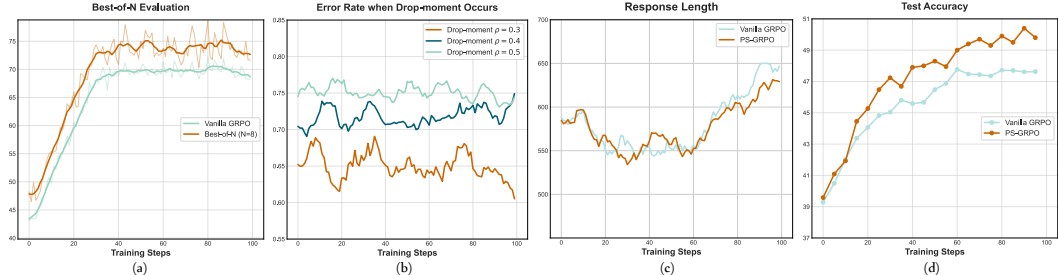

Figure 5: Figure (a) shows the BoN evaluation during GRPO training. We select the best rollout using the mean value of process rewards. Figure (b) illustrates the proportion of rollouts where URSA-8B-RM identifies "drop-moment" and the final results are indeed incorrect. Figures (c) and (d) display the response length and test accuracy during PS-GRPO training.

reward, a scalar process reward $r^i_{s,t}$ is assigned to the $i$-th rollout's $t$-th step. We observe two highly significant conclusions from Figure 4: **(i)** *High susceptibility to reward hacking*. The test accuracy of both variants is lower than vanilla GRPO. This indicates that when process scalar rewards are employed as learning objectives, the model quickly learns strategies that cater to process correctness. However, correctness in the process does not necessarily correlate fully with the heuristics leading to the ground-truth. **(ii)** *PRM's length bias in rewarding*. We observe a trend where increased training leads to shorter model responses and fewer reasoning steps. This phenomenon stems from an inherent length bias in the PRM's training labels; for examples with incorrect answers, steps taken after the first error are unlikely to yield a correct solution. This results in the PRM conservatively rewards the later stages of a reasoning rollout, thereby encouraging the MLLM towards more passive reasoning and a reliance on pattern recognition from existing conditions or simpler heuristics.

**PS-GRPO** The findings above confirm the consideration that flaws in the reward function are amplified when scalar process rewards serve as the optimization target [55, 33]. We ask "Which internal signals of PRM can be trusted?" We employ two views to investigate the reliable region of the PRM: first, the BoN performance during online learning, and second, the PRM's error identification capability. Regarding the latter, we introduce the concept of a *"drop-moment"* within the PRM's reward sequence, which signifies that the PRM questions the validity of the preceding steps. Specifically, for a given solution's PRM reward sequence $\{r^i_{p1}, r^i_{p2}, \cdots, r^i_{pN}\}$, a significant decrease in reward between consecutive steps indicates the occurrence of such a drop-moment.

$$\delta^i_p = \max\left\{\frac{r^i_{p,j} - r^i_{p,j+1}}{r^i_{p,j}}\bigg| j = 0, 1, \ldots, N-1\right\} > \rho \tag{5}$$

Here, $\rho$ represents PRM's drop-moment threshold. As illustrated in Figure 5, the PRM's ability for BoN selection and error identification remains largely unimpaired during the online RL process, exhibiting stable performance. This suggests that *although the scalar reward from the PRM in online RL might be unreliable, the relative quality of solutions it reveals is comparatively trustworthy*.

We leverage this beneficial property to address the reward sparsity problem in GRPO [56–58], aiming to make online RL focus more on learning from rollouts that have accurate results and rigorous processes. We use $\rho$ from Equation 5 as the occurrence threshold for a "drop-moment"; when it occurs, we apply a reward penalty $\gamma$ to rollouts with correct results. This both differentiates the learning value of outcome-correct rollouts and, due to its focus on relative drops in reward sequences, circumvents the impact of PRM's length bias in rewarding.

$$R^i = \begin{cases} 1, & o^i \text{ is correct and } \delta^i_p < \rho \\ 1-\gamma, & o^i \text{ is correct and } \delta^i_p \geq \rho \\ 0, & \text{otherwise} \end{cases} \tag{6}$$

We utilize reward modeling in Equation 6 to conduct a process-supervised GRPO, which facilitates the computation of in-group advantages in Equation 7.

Table 1: Performance Comparison on 6 math reasoning benchmarks. We use accuracy for MathVerse, MathVision, MathVista and GeoQA. We use Score (Loose) on WE-MATH. And average-case accuracy is employed on DYNAMATH. Best results of Closed-source MLLMs are highlighted in green. Best and runner-up results of Open-source MLLMs are highlighted in red and blue.

| | Size | Avg | MathVerse testmini | MathVision full set | MathVista gps | WE-MATH testmini | DYNAMATH testmini | GeoQA full set |
|---|---|---|---|---|---|---|---|---|
| *Closed-Source MLLMs* | | | | | | | | |
| GPT-4o [59] | - | 55.5 | 50.2 | 30.4 | 64.7 | 62.8 | 64.9 | 62.1 |
| GPT-4o-mini [59] | - | 49.2 | 42.3 | 22.8 | 59.9 | 56.3 | 53.5 | 60.1 |
| Gemini-1.5-pro [60] | - | 53.2 | 35.3 | 19.2 | 81.7 | 66.9 | 60.5 | 55.5 |
| *Open-Source General MLLMs* | | | | | | | | |
| InternVL-Chat-V1.5 [61] | 26B | 33.6 | 26.1 | 15.4 | 56.9 | 32.7 | 36.7 | 33.5 |
| Llama-3.2-11B-Vision-Instruct [62] | 11B | 28.0 | 28.9 | 16.9 | 40.9 | 12.0 | 32.2 | 36.9 |
| Qwen2-VL [63] | 8B | 40.2 | 33.6 | 19.2 | 51.0 | 43.0 | 42.1 | 52.2 |
| InternVL2-8B [64] | 8B | 41.8 | 37.0 | 18.4 | 57.7 | 44.9 | 39.7 | 52.8 |
| InternVL2-8B-MPO [65] | 8B | 45.1 | 38.2 | 22.3 | 69.2 | 44.4 | 40.5 | 55.9 |
| InternVL2.5-8B [66] | 8B | 45.2 | 39.5 | 19.7 | 64.9 | 44.7 | 40.5 | 61.6 |
| LLaVA-OneVision [35] | 8B | 40.9 | 28.9 | 18.3 | 71.6 | 44.9 | 37.5 | 43.9 |
| Points-Qwen2.5-Instruct [67] | 8B | 49.8 | 41.1 | 23.9 | 76.0 | 51.0 | 42.8 | 63.8 |
| Gemma3-12B [68] | 12B | 49.8 | 40.1 | 29.1 | 63.6 | 51.7 | 45.8 | 67.7 |
| *Open-Source Reasoning MLLMs* | | | | | | | | |
| Math-LLaVA [15] | 13B | 35.2 | 22.9 | 15.7 | 57.7 | 31.3 | 35.5 | 48.1 |
| MathPUMA-Qwen2-7B [11] | 8B | 39.6 | 33.6 | 14.0 | 48.1 | 41.0 | 37.3 | 63.6 |
| MultiMath [23] | 7B | 43.1 | 27.7 | 16.3 | 66.8 | 42.2 | 37.9 | 67.7 |
| MAVIS [19] | 7B | 44.4 | 35.2 | 18.5 | 64.1 | 44.3 | 36.2 | 68.3 |
| InfiMM-Math [14] | 7B | 48.6 | 40.5 | 18.8 | 77.3 | 48.3 | 38.2 | 68.3 |
| AtomThink-EMOVA [12] | 8B | 49.5 | 42.5 | 24.9 | 75.9 | 49.3 | 40.9 | 63.8 |
| MathGLM-Vision [9] | 9B | 47.6 | 44.2 | 19.2 | 64.2 | 45.2 | 42.2 | 70.4 |
| LlamaV-o1 [69] | 11B | 38.4 | 33.9 | 17.9 | 53.3 | 42.6 | 34.7 | 43.1 |
| Adora-7B [70] | 7B | 54.6 | 50.1 | 25.5 | 77.6 | 58.4 | 45.9 | 70.1 |
| MM-Eureka-7B [71] | 7B | 55.7 | 50.3 | 26.9 | 76.5 | 60.2 | 49.1 | 70.9 |
| OpenVLThinker [72] | 7B | - | 47.9 | 25.3 | 76.4 | - | - | - |
| R1-Onevision [73] | 7B | - | 47.4 | 26.9 | 72.4 | 51.4 | - | - |
| URSA-8B | 8B | 54.7 | 45.7 | 28.7 | 81.7 | 53.6 | 44.7 | 73.5 |
| URSA-8B-PS-GRPO | 8B | 58.2 | 50.9 | 31.5 | 83.2 | 60.7 | 47.4 | 75.6 |

Table 2: Comparison of TTS on URSA-8B and AtomThink-EMOVA using BoN performance.

| Model | Method | MathVerse | | | | MathVista-GPS | | | | MathVision | | | |
|---|---|---|---|---|---|---|---|---|---|---|---|---|---|
| | | N=4 | N=8 | N=16 | N=32 | N=4 | N=8 | N=16 | N=32 | N=4 | N=8 | N=16 | N=32 |
| URSA-8B | Self-Consistency | 49.3 | 50.1 | 50.7 | 50.7 | 82.7 | 83.9 | 84.8 | 85.4 | 29.4 | 31.9 | 32.8 | 33.1 |
| | InternVL2.5-8B ORM | 48.6 | 50.9 | 51.8 | 51.3 | 82.5 | 83.3 | 84.3 | 85.1 | 29.9 | 32.1 | 32.8 | 33.5 |
| | URSA-8B-RM | 53.3 | 54.2 | 54.7 | 55.0 | 83.2 | 85.5 | 86.5 | 87.2 | 31.6 | 33.1 | 34.0 | 35.1 |
| AtomThink-EMOVA | Self-Consistency | 45.9 | 46.7 | 47.1 | 47.3 | 76.8 | 77.9 | 78.6 | 79.0 | 25.3 | 26.8 | 27.6 | 28.0 |
| | InternVL2.5-8B ORM | 45.7 | 45.6 | 46.4 | 46.1 | 76.6 | 77.7 | 78.3 | 79.2 | 26.0 | 26.6 | 27.2 | 27.8 |
| | URSA-8B-RM | 48.0 | 48.8 | 49.3 | 49.6 | 78.0 | 79.6 | 80.5 | 81.0 | 27.5 | 29.0 | 30.2 | 31.0 |

# 5 Experiments

## 5.1 Experimental Setup

**Benchmarks** We evaluate our URSA-series models on 6 widely used reasoning benchmarks, including MathVerse [74], DYNAMATH [75], MathVista [76], WE-MATH [77], GeoQA [40] and MathVision [43]. Detailed description and evaluation criteria can be found in Appendix G.3. We consistently employ zero-shot inference for comparison.

**Baselines** We include some leading proprietary MLLMs, such as GPT-4o and GPT-4o-mini [59]. For open-source MLLMs with comparable size, we select InternVL-series [64, 78], LLaVA-OneVision [35], Gemma3-12B [68], Qwen2-VL [63], and so on. For MLLMs intended for math reasoning purposes, we select AtomThink [12], InfiMM-Math [14], MAVIS [19], MathGLM-Vision [9], LlamaV-o1 [69]. This kind of work focuses on the synthesis of STEM reasoning data or o1-like slow thinking. We do not select baselines that use MathVision as training set for fairness, such as Mulberry-Qwen2-VL-7B [79] and MAmooTH-VL [80]. For PRM's TTS performance, we select Self-Consistency [81] and open-source MLLM as ORM for comparison, such as InternVL2.5-8B [64].

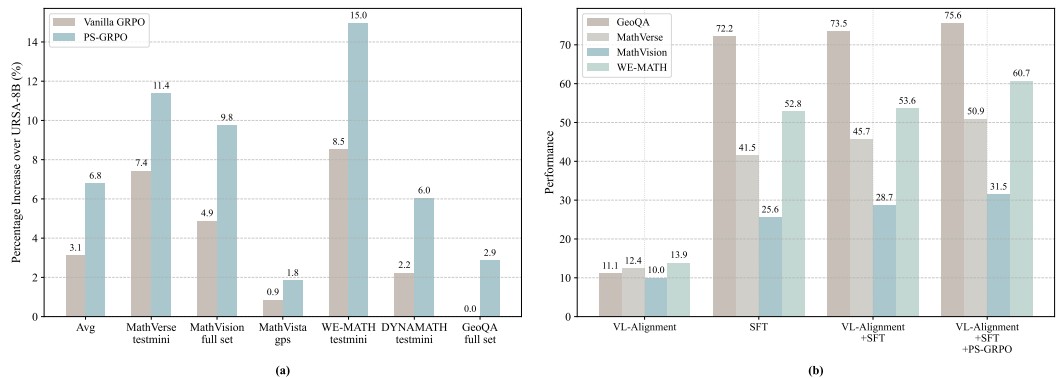

Figure 6: Figure(a) represents the comparison of relative improvements on URSA-8B; Figure(b) illustrates how each training stage contributes to the total performance.

Table 3: Ablation study on DualMath-1.1M (BoN evaluation). w/o $\mathcal{S}_{MIE}$ and w/o $\mathcal{S}_{BEL}$ represents dropping one part of DualMath-1.1M to train the PRM.

| Model | Dataset | MathVerse | | | | MathVista-GPS | | | | MathVision | | | |
|---|---|---|---|---|---|---|---|---|---|---|---|---|---|
| | | N=4 | N=8 | N=16 | N=32 | N=4 | N=8 | N=16 | N=32 | N=4 | N=8 | N=16 | N=32 |
| URSA-8B | DualMath-1.1M | **53.3** | **54.2** | **54.7** | **55.0** | **83.2** | **85.5** | **86.5** | **87.2** | **31.6** | **33.1** | **34.0** | **35.1** |
| | w/o $\mathcal{S}_{MIE}$ | 52.8 | 52.6 | 52.4 | 53.9 | 81.3 | 83.8 | 83.1 | 83.2 | 29.9 | 30.5 | 33.1 | 34.5 |
| | w/o $\mathcal{S}_{BEL}$ | 50.3 | 51.4 | 51.8 | 53.0 | 80.1 | 83.1 | 82.2 | 83.0 | 28.7 | 29.8 | 32.3 | 34.2 |
| AtomThink-EMOVA | DualMath-1.1M | **48.0** | **48.8** | **49.3** | **49.6** | **78.0** | **79.6** | **80.5** | **81.0** | **27.5** | **29.0** | **30.2** | **31.0** |
| | w/o $\mathcal{S}_{MIE}$ | 47.5 | 48.2 | 47.8 | 48.0 | 76.8 | 78.3 | 79.1 | 79.5 | 26.0 | 27.4 | 28.5 | 29.2 |
| | w/o $\mathcal{S}_{BEL}$ | 46.8 | 47.5 | 47.9 | 47.3 | 76.0 | 77.5 | 78.3 | 78.7 | 25.4 | 26.7 | 27.8 | 28.5 |

**Implementation Details**    URSA uses SAM-B+SigLIP-L as the hybrid vision encoder and Qwen2.5-Math-Instruct as the LLM backbone. We employ a two-layer MLP connection for vision-language alignment training. We select 15K data in MMathCoT-1M for PS-GRPO. $\gamma$ and $\rho$ in Equation 6 are set to 0.5 and 0.3, respectively. Details on module selection, data selection, hyperparameters, and time cost are placed in the Appendix D and G.

## 5.2   Main Results

**SoTA Performance**    In Table 1, we present the performance of URSA-8B and URSA-8B-PS-GRPO. First, URSA-8B provides a stronger reasoning foundation model. It demonstrates a 5.2 point advantage over AtomThink-EMOVA which focuses on "slow thinking" training. It also outperforms leading general-purpose MLLMs of comparable size, such as Gemma3-12B and InternVL2.5-8B. URSA-8B-PS-GRPO outperforms GPT-4o across 6 benchmarks on average and shows significant advantages on MathVista-GPS (83.2 vs 62.6), GeoQA (73.5 vs 62.1), and achieves the first surpassing performance on MathVision (31.5 vs 30.4). However, a significant performance gap on DynaMath suggests that smaller-scale MLLMs still lack more robust problem-solving capabilities. Compared to the leading math reasoning MLLM AtomThink-EMOVA-8B and general-purpose MLLM Gemma3-12B in terms of average performance, our model shows advantages of **8.5%** and **8.2%**, respectively. Compared with recent R1-inspired method OpenVLThinker [72] and R1-Onevision [73], we still show significant advantage on MathVision and WE-MATH.

**Effective Best-of-N Evaluation**    In Table 2, we demonstrate the advantages of URSA-8B-RM compared to self-consistency and the ORM baseline on serving TTS [43, 42]. We find that self-consistency remains a strong baseline, which InternVL2.5-8B (serving as the ORM) does not consistently surpasses. However, URSA-8B-RM exhibits more effective BoN evaluation and demonstrates its generalization on AtomThink-EMOVA-8B. In addition, using URSA-8B-RM as the verifier, only 4 samplings can achieve a huge improvement based on URSA-8B. Specifically, it provides a 16.6% and 10.1% relative improvement on MathVerse and MathVision. In Best-of-32 setting, URSA-8B achieve 35.1 and 55.0 in MathVision and MathVerse, showing clear advantage with GPT-4o.

Table 4: Sensitivity analysis on reward penalty and PRM's "drop-moment" judgment.

| $\gamma$ | $\rho$ | MathVerse testmini | MathVision full set | MathVista gps | WE-MATH testmini | DYNAMATH testmini | GeoQA full set | Avg |
|---|---|---|---|---|---|---|---|---|
| 0.5 | 0.3 | 50.9 | 31.5 | **83.2** | 60.7 | **47.4** | **75.6** | **58.2** |
| 0.5 | 0.4 | 49.9 | 30.8 | 81.2 | 59.9 | 46.9 | 75.0 | 57.3 |
| 0.5 | 0.2 | 49.6 | 30.5 | 80.9 | 59.6 | 46.6 | 74.7 | 57.0 |
| 1.0 | 0.3 | 49.0 | 29.4 | 79.8 | 58.8 | 45.3 | 72.5 | 56.3 |
| 0.7 | 0.3 | **52.0** | 31.1 | 81.7 | 59.6 | 47.0 | 73.8 | 57.5 |
| 0.3 | 0.3 | 51.5 | **32.0** | 82.1 | **61.0** | 46.3 | 74.6 | 57.9 |

**PS-GRPO vs Vanilla GRPO**  As shown in Figure 6 (a), given the same training data, hyperparameters, and rollout number PS-GRPO achieves a higher improvement on average performance (6.8% vs 3.1%). PS-GRPO demonstrates an improvement that is nearly double that of vanilla GRPO in WE-MATH and more challenging MathVision, suggesting its effectiveness. We notice that the improvement of RL on MathVista-GPS and GeoQA is relatively small. This is because URSA-8B's inherent abilities have already achieved an effect close to the upper bound on these two benchmarks. However, PS-GRPO still has advantages over vanilla GRPO.

# 6  Analysis

## 6.1  How Each Stage Contributes the Performance

In this section, we demonstrate how each stage contributes to the performance. As demonstrated in Figure 6 (b), all stages make a performance contribution. MMathCoT-1M contributes the highest absolute performance gain. The effect of Alignment-860K is more evident on MathVerse and MathVision, likely because the question images in these two datasets contain richer textual modality information, allowing alignment resources (such as textual images) to better supplement this comprehension capability. PS-GRPO, on the other hand, is dedicated to breaking the bottleneck after large-scale SFT, performing more prominently on WE-MATH and MathVerse with relative improvements of 13.2% and 11.4% respectively, compared to URSA-8B. We provide a generalization validation on InternVL2.5-8B and Multimath in Appendix C.4.

## 6.2  Ablation Studies on Automatic Process Labeling

We give an ablation study on how two parts of DualMath-1.1M contribute to URSA-8B-RM. As shown in Table 3, we can see that the method based on BEL, which focuses on the potential to correctness, and the method based on MIE, which focuses on the perception consistency, both contribute positively to the outcome. This further illustrates that in the process of multimodal math reasoning, image-text inconsistency is widespread and needs to be mitigated. We address this issue by augmenting the process supervision training data through the enforced imposition of common hallucination categories. Specifically, the data generated by BEL demonstrates a more significant impact, indicating that the quality of synthesized data can still be improved.

## 6.3  Sensitivity Analysis on Reward Penalty and Drop-moment

In this section, we conduct a sensitivity analysis on two hyperparameters of PS-GRPO, $\gamma$ and $\rho$. These respectively define the magnitude of the reward penalty for rollouts exhibiting a "drop-moment" and the tolerance threshold for identifying such "drop-moments". As shown in Table 4, our core findings are twofold: (i) The value of $\gamma$ should not be set too high, as this implies excessive trust in the PRM, which may cause the rewards of a group to vanish and lead to training instability. When fixing $\rho$ at 0.3, we find that setting $\gamma$ to a value within a certain appropriate range (we test 0.3-0.7) is generally beneficial for average performance. (ii) An excessively large $\rho$ diminishes reward differentiation, causing the RL behavior to approximate that of vanilla GRPO. Conversely, an excessively small $\rho$ is unreasonable by design, as it is overly sensitive to process reward changes and tends to result in an overly broad range of penalties. In an extreme case where all correct rollouts are penalized, PS-GRPO degenerates back to vanilla GRPO.

Table 5: Comparison with process reward-integrated online RL methods.

| Method | Avg | MathVerse testmini | MathVision full set | MathVista gps | WE-MATH testmini | DYNAMATH testmini | GeoQA full set |
|--------|-----|----------|-----------|----------|----------|----------|----------|
| Variant1 | 55.3 | 48.2 | 27.6 | 79.5 | 57.3 | 45.8 | 72.7 |
| Variant2 | 56.0 | 49.1 | 28.5 | 80.0 | 57.6 | 46.2 | 74.3 |
| DS-GRPO | 56.5 | 49.9 | 29.2 | 79.8 | 58.5 | **47.8** | 74.3 |
| PS-GRPO (Ours) | **58.2** | **50.9** | **31.5** | **83.2** | **60.7** | 47.4 | **75.6** |

## 6.4 PS-GRPO vs. PRM-integrated Baselines

In this section, we further evaluate the two enhanced baselines that utilize scalar process rewards, as previously proposed, alongside the online reinforcement learning method with process-reward participation introduced by DeepSeek (hereafter referred to as DS-GRPO). Specifically, DS-GRPO computes step-level advantages by aggregating all in-batch process rewards: the advantage assigned to each token is the cumulative sum of rewards from all subsequent steps. All these methods represent variants that leverage scalar process rewards to facilitate online reinforcement learning. As shown in the comparison results in the Table 5, PS-GRPO outperforms three strong baselines, demonstrating that our proposed process-to-outcome reward modeling constitutes a leading approach to utilizing process rewards. This further underscores its advantages in mitigating process reward hacking and rewarding length bias.

## 7 Conclusion

In this study, we take the first step to thoroughly explore the application of PRM in multimodal math reasoning. We introduce a three-stage training pipeline URSA designed to address three major challenges. Initially, we provide a large-scale CoT reasoning dataset MMathCoT-1M. This dataset forms the basis for developing URSA-8B, a MLLM with enhanced reasoning capabilities, and paves the way for further TTS or RL scenarios. Next, we present a dual-view automated process supervision annotation method, covering logical validity and perceptual consistency in multimodal scenarios. We introduce the first large-scale process supervision dataset in multimodal reasoning, DualMath-1.1M. Finally, we address reward hacking and rewarding length bias through process-as-outcome modeling, and put forward PS-GRPO, which is a PRM-aided online RL method that surpasses GRPO. The resulting URSA-8B-PS-GRPO model demonstrates superior average performance over leading open-source MLLM such as Gemma3-12B (8.4%) and proprietary GPT-4o (2.7%).

## 8 Acknowledgments

This work was partly supported by the National Natural Science Foundation of China (Grant No. 62576191) and the Shenzhen Science and Technology Program (JCYJ20220818101014030).

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

# Appendices Content

# A Related Work

**Multimodal Mathematical Reasoning**  The mathematical and scientific reasoning capabilities of MLLMs have recently attracted significant attention [11, 82, 35, 83, 84, 5, 14, 80, 85, 86]. Unlike traditional mathematical reasoning tasks in Language Models (LLMs) [1, 87], multimodal mathematical reasoning requires MLLMs to interpret visual information and perform cross-modal reasoning between images and text. Tasks such as solving geometric problems and analyzing graphs are particularly challenging [40]. Recent advances have focused on improving visual mathematical input specialized encoders in specific scenarios [19, 88, 61]. A significant emphasis has also been placed on synthesizing diverse and complex training data. For instance, Math-LLaVA [15] introduces the MathV360K dataset, which categorizes images by complexity and enhances associated questions. Multimath [23] curates high-quality reasoning data from K-12 textbooks and employs GPT-4 for CoT data generation and validation. R-CoT [17] further diversifies problems through a two-stage reverse question-answer generation process. These data synthesis methods are widely adopted in academia and industry due to their demonstrated efficiency [89–92]. PeRL [93] explores RL for interleaved multimodal reasoning.

**Process Reward Model**  Recent studies have explored test-time scaling laws in LLMs, aiming to identify optimal reasoning ecectories from diverse thinking trajectories [26, 94–96, 31, 97, 98]. Initial efforts, such as self-consistency [81], have laid the groundwork for test-time scaling. OpenAI has introduced verifiers to supervise and select reasoning paths during inference [41]. Math-Shepherd [43] evaluates intermediate reasoning steps based on their likelihood of leading to correct answers, while OmegaPRM [42] constructs PRM training data and employs MCTS for training. Despite these advancements, the lack of models with robust CoT reasoning capabilities and limited exploration into diverse reward model training data remain significant bottlenecks in multimodal mathematical reasoning. Some concurrent work also begins to pay attention to PRM-assisted visual reasoning, such as construction and benchmarking [99–101].

# B Preliminary

## B.1 Group Relative Policy Optimization

Vanilla GRPO eliminates value function in PPO and estimates the advantages within online rollout group. Given a question with image $q$ and ground-truth $y$, policy model $\pi_{\theta_{old}}$ samples a group of $G$ responses $\{o^i\}_{i=1}^G$. GRPO compute the $i$-th response's advantage through normalizing in-group rewards $\{r^j\}_{j=1}^G$, and employs PPO's clipped objective and KL penalty term:

$$A^i = \frac{r^i - \text{mean}(\{r^j\}_{j=1}^G)}{\text{std}(\{r^j\}_{j=1}^G)} \tag{7}$$

$$\mathcal{J}_{GRPO}(\theta) = \mathbb{E}_{(q,y)\sim\mathcal{D},\{o^i\}_{i=1}^G \sim \pi_{\theta_{old}}(\cdot|q)}$$
$$\left[\frac{1}{G}\sum_{i=1}^G \frac{1}{|o^i|}\sum_{t=1}^{|o^i|}(\min(r_t^i(\theta)A^i, \text{clip}(r_t^i(\theta), 1-\epsilon, 1+\epsilon)A^i) - \beta D_{KL}^{i,t}(\pi_\theta||\pi_{ref}))\right] \tag{8}$$

We introduce the two variants of PRM-integrated GRPO discussed in Section 4. Given PRM $\mathcal{M}_p$ and process reward sequences $r_s = \mathcal{M}_p(\{s_1, s_2, \cdots, s_N\})$ (i) *Variant 1*: Given verifiable outcome reward $r_o^i$, we set a single rollout's reward as $r^i = r_o^i + r_s^i$. (ii) *Variant 2*: We utilize a step-level reward and a multiple relative advantage calculated by the mean value of process rewards from each rollout:

$$A_t^i = r_{s,t}^i \underbrace{\frac{\bar{r_s^i} - \text{mean}(\{\bar{r_s^j}\}_{j=1}^G)}{\text{std}(\{\bar{r_s^j}\}_{j=1}^G)}}_{\text{GRPO with process rewards}} + \underbrace{\frac{r_o^i - \text{mean}(\{r_o^j\}_{j=1}^G)}{\text{std}(\{r_o^j\}_{j=1}^G)}}_{\text{GRPO with outcome rewards}} \tag{9}$$

in which $\bar{r_s^i} = \text{mean}(\mathcal{M}_p(\{s_1^i, s_2^i, \cdots, s_{T_i}^i\}))$.

## B.2 Test-Time Scaling by Best-of-N evaluation

Following previous works [41, 43], we adopt BoN evaluation for TTS. Given $N$ response samplings for a question $q$. The PRM is used to give process reward for each sampling. We use mean value of process rewards to select the best single sampling:

$$a_{\text{prm}} = \arg \max_{s_i} \text{mean}\{\mathcal{M}_p(q, s_i)\} \tag{10}$$

Some other works [102] merge self-consistency and PRM to employ a voting-based score cumulation. But we don't select this method for a simpler evaluation manner.

# C Supplementary Results

## C.1 Fine-grained Comparison on Used benchmarks

In this section, we provide some fine-grained results for a clearer comparison. As demonstrated in Table 6, our proposed methods demonstrate significant advantages. Compared to closed-source models like GPT-4o and GPT-4V, our URSA-8B and URSA-8B-PS-GRPO show strong competitiveness. Among open-source models, the performance improvements are even more evident. Our URSA-8B model outperforms other open-source models such as InternLM-XComposer2-VL and Ovis1.6-Gemma2-9B in most subtasks. When combined with PS-GRPO, the URSA-8B-PS-GRPO model achieves even better results, showing significant improvements in subtasks like Alg, AnaG, CombG, and others. Our methods particularly excel in complex mathematical reasoning tasks, demonstrating their powerful mathematical reasoning capabilities. These results highlight the effectiveness of our proposed MMathCoT-1M and PS-GRPO methods in enhancing the mathematical reasoning abilities of models, especially in visual mathematical problems.

In Dynamath (Table 7), compared to open-source MLLMs, the URSA series has obvious advantages in plane geometry and algebra. Surprisingly, from the knowledge level classification, the URSA series model performs excellently at the undergraduate level, which is partly attributable to its math-intensive alignment and large-scale instruction fine-tuning.

In MathVerse (Table 8), we can see that URSA series model marginally surpass GPT-4o on average. Besides, compared with other open-source MLLMs, URSA-8B-PS-GRPO outperforms leading AtomThink-EMOVA-8B and InternVL2.5-8B with **8.4** and **11.4** points. In WE-MATH 9, URSA-

Table 6: Performance comparison of different MLLMs on MathVision.

| Model | Size | ALL | Alg | AnaG | Ari | CombG | Comb | Cnt | DescG | GrphT | Log | Angle | Area | Len | SolG | Stat | Topo | TransG |
|---|---|---|---|---|---|---|---|---|---|---|---|---|---|---|---|---|---|---|
| | | | | | | | *Baselines* | | | | | | | | | | | |
| Human | - | 68.8 | 55.1 | 78.6 | 99.6 | 98.4 | 43.5 | 98.5 | 91.3 | 62.2 | 61.3 | 33.5 | 47.2 | 73.5 | 87.3 | 93.1 | 99.8 | 69.0 |
| | | | | | | | *Closed-source MLLMs* | | | | | | | | | | | |
| GPT-4o | - | 30.4 | 42.0 | 39.3 | 49.3 | 28.9 | 25.6 | 22.4 | 24.0 | 23.3 | 29.4 | 17.3 | 29.8 | 30.1 | 29.1 | 44.8 | 34.8 | 17.9 |
| GPT-4V | - | 22.8 | 27.3 | 32.1 | 35.7 | 21.1 | 16.7 | 13.4 | 22.1 | 14.4 | 16.8 | 22.0 | 22.2 | 20.9 | 23.8 | 24.1 | 21.7 | 25.6 |
| CoT GPT-4V | - | 24.0 | 26.7 | 26.2 | 38.6 | 22.1 | 24.4 | 19.4 | 27.9 | 23.3 | 25.2 | 17.3 | 21.4 | 23.4 | 23.8 | 25.9 | 4.4 | 25.6 |
| Gemini-1.5-Pro | - | 19.2 | 20.3 | 35.7 | 34.3 | 19.8 | 15.5 | 20.9 | 26.0 | 26.7 | 22.7 | 14.5 | 14.4 | 16.5 | 18.9 | 10.3 | 26.1 | 17.3 |
| | | | | | | | *Open-source MLLMs* | | | | | | | | | | | |
| LLaVA-1.5 | 7B | 8.5 | 7.0 | 7.1 | 10.7 | 7.1 | 4.8 | 10.5 | 7.7 | 10.0 | 9.2 | 15.6 | 10.2 | 9.8 | 5.3 | 8.6 | 4.4 | 4.8 |
| LLaVA-1.5 | 13B | 11.1 | 7.0 | 14.3 | 14.3 | 9.1 | 6.6 | 6.0 | 13.5 | 5.6 | 13.5 | 10.4 | 12.6 | 14.7 | 11.5 | 13.8 | 13.0 | 10.7 |
| InternLM-XComposer2-VL | 7B | 14.5 | 9.3 | 15.5 | 12.1 | 15.3 | 11.3 | 10.5 | 14.4 | 22.2 | 19.3 | 19.7 | 15.6 | 15.0 | 11.9 | 15.5 | 26.1 | 15.5 |
| Ovis1.6-Gemma2-9B | 9B | 18.8 | 13.3 | 15.5 | 22.1 | 17.9 | 11.3 | 22.4 | 23.1 | 20.0 | 20.2 | 20.8 | 18.0 | 24.7 | 15.6 | 20.7 | 17.4 | 20.8 |
| MiniCPM-v2.6 | 8B | 18.4 | 9.9 | 19.0 | 18.6 | 21.8 | 13.1 | 13.4 | 17.3 | 20.0 | 16.0 | 25.4 | 19.4 | 20.7 | 15.2 | 27.6 | 30.4 | 22.0 |
| LLaVA-OneVision | 8B | 18.3 | 11.6 | 16.7 | 20.7 | 18.5 | 11.9 | 14.9 | 19.2 | 13.3 | 20.2 | 17.9 | 21.6 | 23.4 | 12.3 | 22.4 | 13.0 | 24.4 |
| Qwen2-VL | 8B | 19.2 | 15.4 | 20.2 | 19.3 | 16.9 | 16.7 | 17.9 | 22.1 | 22.2 | 16.0 | 19.1 | 22.4 | 22.5 | 14.8 | 19.0 | 4.3 | 23.8 |
| InternVL2-8B | 8B | 18.4 | 18.6 | 22.6 | 28.6 | 22.1 | 13.7 | 10.4 | 11.5 | 13.3 | 21.0 | 20.8 | 22.4 | 20.5 | 16.8 | 17.2 | 26.1 | 24.2 |
| InternVL2.5-8B | 8B | 19.7 | 15.1 | 23.8 | 29.3 | 16.2 | 8.9 | 11.9 | 10.6 | 8.9 | 18.5 | 22.0 | 19.4 | 15.4 | 13.9 | 22.4 | 21.7 | 19.6 |
| | | | | | | | *Open-source Math MLLMs* | | | | | | | | | | | |
| Math-LLaVA | 13B | 15.7 | 9.0 | 20.2 | 15.7 | 18.2 | 10.1 | 10.5 | 16.4 | 14.4 | 16.0 | 20.2 | 18.4 | 17.6 | 9.4 | 24.1 | 21.7 | 17.9 |
| Multimath | 7B | 16.3 | 11.3 | 21.1 | 15.5 | 15.9 | 11.3 | 12.1 | 15.5 | 15.9 | 18.5 | 20.1 | 16.4 | 21.3 | 13.3 | 14.6 | 13.3 | 20.8 |
| Math-PUMA-Qwen2-7B | 8B | 14.0 | 5.0 | 21.1 | 21.1 | 21.1 | 11.3 | 5.6 | 15.7 | 10.5 | 13.8 | 11.7 | 15.8 | 12.2 | 17.8 | 19.2 | 15.8 | 12.2 |
| MAVIS | 7B | 18.5 | 17.5 | 19.5 | 21.5 | 19.0 | 12.0 | 14.0 | 18.0 | 16.0 | 19.0 | 21.0 | 18.5 | 19.5 | 15.0 | 19.0 | 20.0 | 20.0 |
| AtomThink-EMOVA | 8B | 24.9 | 23.5 | 25.5 | 32.0 | 21.0 | 15.8 | 19.5 | 21.5 | 22.5 | 21.5 | 26.5 | 25.5 | 26.5 | 27.5 | 28.0 | 23.0 | 22.5 |
| URSA-8B | 8B | 28.7 | 28.1 | 26.2 | 35.0 | 22.1 | 15.5 | 19.4 | 18.3 | 22.2 | 21.8 | 37.0 | 27.0 | 26.5 | 31.1 | 27.6 | 17.4 | 23.8 |
| URSA-8B-PS-GRPO | 8B | 31.5 | 30.1 | 28.6 | 29.3 | 31.5 | 20.8 | 20.9 | 26.9 | 17.8 | 24.4 | 35.8 | 33.6 | 37.2 | 37.7 | 25.9 | 26.1 | 35.1 |

series outperforms leading general-purpose and math reasoning MLLMs in three-stage accuracy. Also, the URSA series has remarkable strengths in solid figures, transformations, positions, and directions. This is mainly due to large-scale alignment and instruction tuning, which builds its foundation in understanding mathematical elements.

Table 7: Detailed performance comparison of MLLMs on **DYNAMATH** *testmini* dataset, broken down by subject area and knowledge level.

| Model | Size | ALL | PG | SG | AG | AL | PT | GT | AR | Elem. | High | Undergrad. |
|---|---|---|---|---|---|---|---|---|---|---|---|---|
| *Closed-source MLLMs* | | | | | | | | | | | | |
| GPT-4o | - | **64.9** | 56.8 | 52.0 | 61.0 | 76.9 | 51.8 | 58.1 | 61.5 | 68.6 | 61.8 | 36.8 |
| Claude-3.5-Sonnet | - | 64.8 | 49.9 | 49.3 | 55.3 | 81.0 | 44.1 | 69.4 | 61.2 | 66.7 | 62.6 | 33.3 |
| Gemini-1.5-Pro | - | 60.5 | 52.7 | 42.7 | 61.6 | 70.8 | 20.6 | 65.2 | 54.2 | 62.9 | 59.2 | 37.1 |
| *Open-source MLLMs* | | | | | | | | | | | | |
| Llava-v1.5-7B | 7B | 16.6 | 10.5 | 7.3 | 19.5 | 6.5 | 8.2 | 32.3 | 10.8 | 18.9 | 13.3 | 11.7 |
| Llava-v1.6-34B | 34B | 27.1 | 21.4 | 25.3 | 27.6 | 14.9 | 7.6 | 32.7 | 23.1 | 35.9 | 23.8 | 16.6 |
| Deepseek-VL-7B-Chat | 7B | 21.5 | 16.0 | 13.3 | 26.5 | 12.9 | 4.7 | 32.3 | 12.7 | 28.3 | 19.0 | 16.0 |
| InternVL2-8B | 8B | 39.7 | 33.9 | 37.3 | 32.5 | 46.9 | 15.9 | 42.1 | 37.3 | 51.1 | 37.4 | 19.6 |
| Qwen2-VL | 8B | 42.1 | 40.3 | 38.7 | 39.9 | 37.1 | 8.2 | 44.8 | 39.2 | 47.6 | 42.2 | 24.4 |
| AtomThink-EMOVA | 8B | 40.9 | 42.0 | 37.9 | 33.6 | 58.0 | 23.0 | 44.0 | 38.4 | 52.5 | 43.5 | 32.0 |
| URSA-8B | 8B | 44.7 | 48.1 | 38.0 | 33.7 | 66.9 | 24.7 | 39.2 | 38.5 | 53.5 | 44.3 | 41.8 |
| URSA-8B-PS-GRPO | 8B | 47.4 | 49.7 | 40.1 | 35.2 | 65.7 | 24.7 | 45.2 | 41.1 | 53.5 | 46.7 | 43.2 |

Table 8: Comparison with closed-source MLLMs and open-source MLLMs on **MATHVERSE** *testmini*. The best results of Closed-source MLLMs are highlighted. The best and second-best results of Open-source MLLMs are highlighted.

| Model | #Params | ALL | TD | TL | TO | VI | VD | VO |
|---|---|---|---|---|---|---|---|---|
| *Baselines* | | | | | | | | |
| Random | - | 12.4 | 12.4 | 12.4 | 12.4 | 12.4 | 12.4 | 12.4 |
| Human | - | 64.9 | 71.2 | 70.9 | 41.7 | 61.4 | 68.3 | 66.7 |
| *Closed-Source MLLMs* | | | | | | | | |
| GPT-4o | - | 50.8 | 59.8 | 50.3 | 52.4 | 48.0 | 46.5 | 47.6 |
| GPT-4V | - | 39.4 | 54.7 | 41.4 | 48.7 | 34.9 | 34.4 | 31.6 |
| Gemini-1.5-Flash-002 | - | 49.4 | 57.2 | 50.5 | 50.3 | 47.6 | 45.1 | 45.4 |
| Gemini-1.5-Pro | - | 35.3 | 39.8 | 34.7 | 44.5 | 32.0 | 36.8 | 33.3 |
| Claude-3.5-Sonnet | - | - | - | - | - | - | - | - |
| Qwen-VL-Plus | - | 21.3 | 26.0 | 21.2 | 25.2 | 18.5 | 19.1 | 21.8 |
| *Open-Source General MLLMs* | | | | | | | | |
| mPLUG-Owl2-7B | 7B | 10.3 | 11.6 | 11.4 | 13.8 | 11.1 | 9.4 | 8.0 |
| MiniGPT4-7B | 7B | 12.2 | 12.3 | 12.9 | 13.4 | 12.5 | 14.8 | 8.7 |
| LLaVA-1.5-13B | 13B | 12.7 | 17.1 | 12.0 | 22.6 | 12.6 | 12.7 | 9.0 |
| SPHINX-V2-13B | 13B | 16.1 | 20.8 | 14.1 | 14.0 | 16.4 | 15.6 | 16.2 |
| LLaVA-NeXT-34B | 34B | 34.6 | 49.0 | 37.6 | 30.1 | 35.2 | 28.9 | 22.4 |
| InternLM-XComposer2-VL | 7B | 25.9 | 36.9 | 28.3 | 42.5 | 20.1 | 24.4 | 19.8 |
| Deepseek-VL | 8B | 19.3 | 23.0 | 23.2 | 23.1 | 20.2 | 18.4 | 11.8 |
| LLaVA-OneVision (SI) | 8B | 28.9 | 29.0 | 31.5 | 34.5 | 30.1 | 29.5 | 26.9 |
| Qwen2-VL | 8B | 33.6 | 37.4 | 33.5 | 35.0 | 31.3 | 30.3 | 28.1 |
| InternVL2-8B | 8B | 35.9 | 39.0 | 33.8 | 36.0 | 32.2 | 30.9 | 27.7 |
| InternVL2.5-8B | 8B | 39.5 | 43.0 | 43.0 | 43.0 | 43.0 | 42.2 | 22.8 |
| *Open-Source Math MLLMs* | | | | | | | | |
| G-LLaVA-7B | 7B | 16.6 | 20.9 | 20.7 | 21.1 | 17.2 | 14.6 | 9.4 |
| Math-LLaVA-13B | 13B | 22.9 | 27.3 | 24.9 | 27.0 | 24.5 | 21.7 | 16.1 |
| Math-PUMA-Qwen2-7B | 8B | 33.6 | 42.1 | 35.0 | 39.8 | 33.4 | 31.6 | 26.0 |
| Math-PUMA-DeepSeek-Math | 7B | 31.8 | 43.4 | 35.4 | 47.5 | 33.6 | 31.6 | 14.7 |
| MAVIS-7B | 7B | 35.2 | 43.2 | 37.2 | 35.2 | 34.1 | 29.7 | 31.8 |
| InfiMM-Math | 7B | 40.5 | 46.7 | 39.4 | 41.6 | 38.1 | 40.4 | 27.8 |
| Multimath-7B | 7B | 27.7 | 34.8 | 30.8 | 35.3 | 28.1 | 25.9 | 15.0 |
| AtomThink-EMOVA | 8B | 42.5 | 48.1 | 47.7 | 45.7 | 44.0 | 44.2 | 26.8 |
| URSA-8B | 8B | 45.7 | 55.3 | 48.3 | 51.8 | 46.4 | 43.9 | 28.6 |
| URSA-8B-PS-GRPO | 8B | 50.9 | 57.3 | 52.2 | 50.2 | 48.7 | 47.6 | 31.5 |

## C.2 Scaling Law of MMathCoT-1M

To better illustrate the effectiveness of MMathCoT-1M, we examine the scaling laws of SFT by training models on randomly selected samples representing various ratios of the full dataset.

As shown in Table 10, we can see that MMathCoT-1M clearly shows a training time scaling law, further validates the effectiveness of the synthesized data.

Table 9: Accuracy comparison with closed-source MLLMs and open-source MLLMs on **WE-MATH** *testmini* subset. First 3 columns show the overall performance on one-step, two-step and three-step problems. The other columns are used to demonstrate the performance in different problem strategies. Red indicates the best performance and Blue indicates the second best performance among open-source models.

| Model | #Params | S1 | S2 | S3 | Mem | | PF | | SF | | TMF | | PD | | | |
|---|---|---|---|---|---|---|---|---|---|---|---|---|---|---|---|---|
| | | | | | UCU | AL | CPF | UPF | CSF | USF | BTF | CCF | Dir | Pos | RoM | CCP |
| *Closed-source MLLMs* | | | | | | | | | | | | | | | | |
| GPT-4o | - | 72.8 | 58.1 | 43.6 | 86.6 | 39.1 | 77.4 | 71.6 | 84.5 | 62.3 | 58.7 | 69.4 | 93.1 | 72.7 | 47.5 | 73.3 |
| GPT-4V | - | 65.5 | 49.2 | 38.2 | 82.5 | 38.4 | 70.7 | 60.2 | 76.6 | 56.3 | 57.8 | 67.7 | 79.3 | 57.5 | 47.8 | 63.3 |
| Gemini-1.5-Pro | - | 56.1 | 51.4 | 33.9 | 51.0 | 31.2 | 61.8 | 45.0 | 70.0 | 57.5 | 39.2 | 62.7 | 68.8 | 54.1 | 40.7 | 60.0 |
| Qwen-VL-Max | - | 40.8 | 30.3 | 20.6 | 19.4 | 25.3 | 39.8 | 41.4 | 43.6 | 48.0 | 43.8 | 43.4 | 41.4 | 35.1 | 40.7 | 26.7 |
| *Open-source General MLLMs* | | | | | | | | | | | | | | | | |
| LLaVA-1.6 | 7B | 23.0 | 20.8 | 15.8 | 18.5 | 20.5 | 16.9 | 29.6 | 15.6 | 18.6 | 42.7 | 24.1 | 17.6 | 43.3 | 28.9 | 26.7 |
| LLaVA-1.6 | 13B | 29.4 | 25.3 | 32.7 | 21.7 | 23.2 | 23.4 | 34.7 | 25.3 | 26.4 | 37.5 | 41.7 | 26.9 | 28.9 | 37.1 | 30.0 |
| GLM-4V-9B | 9B | 47.3 | 37.2 | 38.2 | 53.4 | 37.0 | 51.3 | 46.5 | 50.6 | 38.2 | 44.1 | 45.2 | 41.0 | 49.3 | 36.8 | 53.3 |
| MiniCPM-LLaMA3-V2.5 | 8B | 39.8 | 31.1 | 29.7 | 28.6 | 37.0 | 40.8 | 39.8 | 41.0 | 38.6 | 32.0 | 42.7 | 41.0 | 42.7 | 44.0 | 43.3 |
| LongVA | 7B | 43.5 | 30.6 | 28.5 | 24.5 | 39.8 | 45.1 | 40.8 | 51.9 | 42.5 | 45.6 | 44.6 | 44.5 | 40.7 | 47.5 | 20.0 |
| InternLM-XComposer2-VL | 7B | 47.0 | 33.1 | 33.3 | 31.3 | 46.5 | 47.7 | 42.6 | 51.4 | 43.9 | 41.1 | 50.6 | 65.5 | 53.9 | 55.2 | 40.0 |
| Phi3-Vision | 4.2B | 42.1 | 34.2 | 27.9 | 28.7 | 16.0 | 47.2 | 38.8 | 50.0 | 44.4 | 28.8 | 31.2 | 48.6 | 49.2 | 26.4 | 50.0 |
| DeepSeek-VL | 7B | 32.6 | 26.7 | 25.5 | 16.6 | 35.1 | 27.3 | 38.0 | 24.2 | 38.7 | 50.0 | 23.3 | 24.5 | 41.0 | 51.7 | 23.3 |
| InternVL2-8B | 8B | 59.4 | 43.6 | 35.2 | 71.4 | 20.5 | 62.0 | 55.5 | 67.1 | 57.3 | 54.0 | 60.5 | 58.6 | 63.6 | 44.5 | 50.0 |
| InternVL2.5-8B | 8B | 58.7 | 43.1 | 38.8 | 48.7 | 35.8 | 65.5 | 54.5 | 62.3 | 61.5 | 47.8 | 60.3 | 79.0 | 64.0 | 51.1 | 63.3 |
| Qwen2-VL | 8B | 59.1 | 43.6 | 26.7 | 62.7 | 37.2 | 62.6 | 60.8 | 65.7 | 49.2 | 52.5 | 49.2 | 48.1 | 68.2 | 55.0 | 56.7 |
| Gemma3-12B | 12B | 64.3 | 47.2 | 42.1 | 83.1 | 33.9 | 70.2 | 58.2 | 77.5 | 61.1 | 50.1 | 63.7 | 82.6 | 58.4 | 36.8 | 60.0 |
| *Open-source Math MLLMs* | | | | | | | | | | | | | | | | |
| G-LLaVA | 7B | 32.4 | 30.6 | 32.7 | 33.3 | 29.1 | 32.0 | 37.9 | 19.6 | 33.5 | 37.1 | 32.8 | 31.2 | 33.2 | 25.6 | 40.0 |
| Math-LLaVA | 13B | 38.7 | 34.2 | 34.6 | 30.3 | 17.9 | 39.2 | 40.4 | 37.1 | 37.7 | 53.0 | 51.3 | 30.8 | 30.8 | 40.9 | 46.7 |
| Math-PUMA-Qwen2-7B | 8B | 53.3 | 39.4 | 36.4 | 63.5 | 42.5 | 60.2 | 45.9 | 66.2 | 48.6 | 42.3 | 53.5 | 31.2 | 37.7 | 40.4 | 46.7 |
| MAVIS w/o DPO | 7B | 56.9 | 37.1 | 33.2 | - | - | - | - | - | - | - | - | - | - | - | - |
| MAVIS | 7B | 57.2 | 37.9 | 34.6 | - | - | - | - | - | - | - | - | - | - | - | - |
| URSA-8B | 8B | 63.1 | 56.4 | 41.8 | 59.1 | 32.5 | 72.3 | 60.3 | 70.9 | 66.0 | 51.4 | 59.8 | 58.3 | 39.5 | 58.8 | 53.3 |
| URSA-8B-PS-GRPO | 8B | 68.6 | 64.2 | 52.7 | 52.6 | 63.5 | 68.5 | 64.1 | 68.8 | 73.6 | 69.4 | 75.8 | 72.1 | 72.6 | 73.6 | 63.3 |

Table 10: Scaling law validation on URSA-8B using different ratios of the MMathCoT-1M.

| Ratio | MathVerse | MathVision | MathVista-GPS | WEMATH | DYNAMATH |
|---|---|---|---|---|---|
| 1/4 | 34.7 | 20.5 | 68.5 | 43.5 | 36.6 |
| 1/2 | 40.5 | 22.8 | 72.3 | 47.7 | 38.8 |
| 3/4 | 42.0 | 26.7 | 77.9 | 50.9 | 42.2 |
| 1 | **45.7** | **28.7** | **81.7** | **53.6** | **44.7** |

## C.3 Higher Upper Bound Taken from Stage I

In Stage I, we obtain a more powerful base MLLM with enhanced reasoning capabilities through math-intensive vision-language alignment and instruction fine-tuning. Beyond the results in Table 1, we explain why Stage I can better serve subsequent experiments, focusing on test-time scaling and PRM applications. We select MathVerse, MathVision, and MathVista-GPS to observe the **pass@N** metric. As demonstrated in Figure 7, we find that URSA-8B consistently outperforms current leading general MLLMs and math reasoning MLLMs. This indicates that while current trends favor RL-related techniques, the scaling law of supervised fine-tuning can still demonstrate its role in breaking through the base model's limitations. This naturally brings advantages in areas such as BoN evaluation and the proportion of valuable rollouts in online RL. First, URSA-8B's higher upper bound leads to richer and more reliable process label generation in Stage II. Furthermore, since recent works claims that RL can only approach the optimal solution within its own exploration path [29, 4, 103], Stage I naturally expands the potential upper limit of the RL stage. This provides the most fundamental advantage to the performance of URSA-PS-GRPO-8B.

## C.4 Generalization Validation

To further vallidate the effectiveness of proposed MMathCoT-1M and PRM aided PS-GRPO. We select InternVL2.5-8B from the general-purpose MLLMs and Multimath from the math reasoning MLLMs for a generalization validation experiment. We do not conduct additional hyperparameter tuning but almost directly adopt the settings from Table 14. The experiment on InternVL2.5-8B and Multimath are implemented on Meng et al. [71] and TRL [104]. Given that these two models

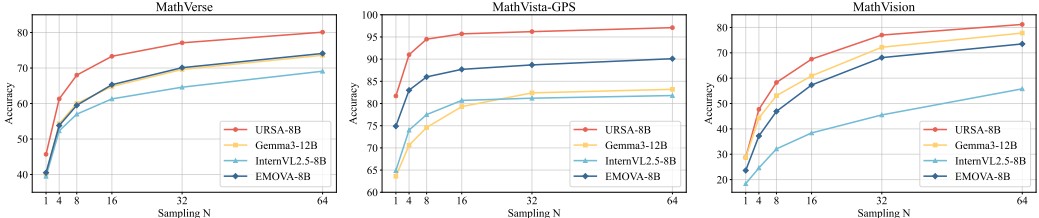

Figure 7: **Pass@N** evaluation on three benchmarks.

Figure 8: The progress of MMathCoT-1.1M and URSA-8B-RM aided PS-GRPO on InternVL2.5-8B and MultiMath.

| Model | Avg | MathVerse testmini | MathVision full set | MathVista gps | WE-MATH testmini | DYNAMATH testmini | GeoQA full set |
|---|---|---|---|---|---|---|---|
| InernVL2.5-8B | 45.2 | 39.5 | 19.7 | 64.9 | 44.7 | 40.5 | 61.6 |
| + MMathCoT-1.1M | 51.7 (↑6.5) | 43.3 (↑3.8) | 25.9 (↑6.2) | 77.9 (↑13.0) | 49.1 (↑4.4) | 45.3 (↑4.8) | 68.8 (↑7.2) |
| + PS-GRPO | **54.7** (↑9.5) | **47.5** (↑8.0) | **28.5** (↑8.8) | **80.1** (↑15.2) | **55.3** (↑10.6) | **45.9** (↑5.4) | **71.1** (↑9.5) |
| MultiMath | 43.1 | 27.7 | 16.3 | 66.8 | 42.2 | 37.9 | 67.7 |
| + MMathCoT-1.1M | 48.7 (↑5.6) | 36.9 (↑9.2) | 22.7 (↑6.4) | 74.4 (↑7.6) | 45.8 (↑3.6) | 40.4 (↑2.5) | 72.2 (↑4.5) |
| + PS-GRPO | **51.2** (↑8.1) | **39.7** (↑12.0) | **24.4** (↑8.1) | **77.7** (↑10.9) | **49.3** (↑7.1) | **42.6** (↑4.7) | **73.5** (↑5.8) |

Table 11: Comparison of TTS with different models using BoN performance on WE-MATH, DYNA-MATH, and GeoQA.

| Model | Method | WE-MATH | | | | DYNAMATH | | | | GeoQA | | | |
|---|---|---|---|---|---|---|---|---|---|---|---|---|---|
| | | N=4 | N=8 | N=16 | N=32 | N=4 | N=8 | N=16 | N=32 | N=4 | N=8 | N=16 | N=32 |
| URSA-8B | Self-Consistency | 56.3 | 57.0 | 57.7 | 58.0 | 46.2 | 46.7 | 47.5 | 48.0 | 74.1 | 75.3 | 75.9 | 75.9 |
| | InternVL2.5-8B ORM | 56.0 | 56.8 | 57.4 | 57.7 | 45.9 | 46.5 | 47.2 | 47.7 | 73.8 | 75.0 | 75.6 | 75.6 |
| | URSA-8B-RM | **58.2** | **59.0** | **59.3** | **59.7** | **47.5** | **48.4** | **49.5** | **50.5** | **76.1** | **77.3** | **78.0** | **78.1** |
| AtomThink-EMOVA | Self-Consistency | 51.7 | 52.4 | 52.9 | 53.6 | 42.3 | 43.0 | 43.7 | 44.0 | 65.7 | 66.5 | 66.6 | 66.8 |
| | InternVL2.5-8B ORM | 51.5 | 52.2 | 52.7 | 53.3 | 42.1 | 42.8 | 43.5 | 43.7 | 65.5 | 66.3 | 66.4 | 66.6 |
| | URSA-8B-RM | **53.7** | **54.5** | **55.0** | **55.8** | **44.1** | **44.9** | **45.6** | **46.0** | **67.9** | **68.8** | **69.0** | **69.3** |

have already undergone sufficient alignment for general domains or specific vertical domains upon their release, we only carry out two stages of training: (i) MMathCoT-1M is used to enhance the base model's mathematical reasoning capabilities; (ii) URSA-8B-RM is involved in the PS-GRPO process. We present the results in Table 8. The proposed MMathCoT-1M and PRM aided PS-GRPO demonstrate remarkable generalization capabilities across different models and benchmarks. When applied to InternVL2.5-8B and MultiMath, both models show significant performance improvements. For InternVL2.5-8B, adding MMathCoT-1M boosts the average score from 45.2 to 51.7, with even more significant gains when combined with PS-GRPO, reaching 54.7. Similarly, for MultiMath, the average score increases from 43.1 to 48.7 with MMathCoT-1M and further to 51.2 with PS-GRPO. These results highlight the effectiveness of our approach in enhancing mathematical reasoning capabilities across diverse models and tasks. The performance improvements are consistent across various benchmarks, including MathVerse, MathVision, MathVista, WE-MATH, DYNAMATH, and GeoQA, indicating that our methods are not only effective but also broadly applicable.

### C.5  Implementary Results on Other Benchmarks

We provide supplementary results on WE-MATH, DYNAMATH and GeoQA when comparing BoN selection. As shown in table 11, URSA-8B-RM remains an advantage with Self-consistency and InternVL2.5-8B ORM. When employing URSA-8B as reasoning model, URSA-8B-RM outperforms Self-consistency with 4.6%, 4.5% and 2.7% relative improvements in Best-of-8 performance.

## D  Module Selection Criteria

As for module selection, we primarily considered the choice of the vision encoder and the LLM backbone.

**Vision Encoder** To train a reasoning model with higher process sensibility and facilitate PRM training, we first conduct captioning tests on open-source models like DeepSeek-VL, Qwen2-VL, etc., using a manually selected dataset (approximately 80 examples). These examples primarily include function-related and geometry problems prone to visual confusion. We manually inspect the outputs of these open-source models and find that Qwen2-VL and LLaVA-OneVision performed poorly; even though their performance on standard benchmarks is good, they fail to ensure sufficiently accurate mathematical descriptions. However, DeepSeekVL's native hybrid vision tower design, integrating high- and low-resolution processing, subjectively exhibit better recognition accuracy. We speculate that this is due to QwenViT [105] being more heavily biased towards general multimodal tasks, resulting in less precise mathematical descriptions compared to simpler vision backbones. Therefore, we choose the SigLiP-L+SAM-B hybrid vision tower design.

**LLM Backbone** Considering the open-source influence of the QwenLM-Series, we follow the choice of prior work such as MathPUMA [11] and Multimath [23] by using the QwenLM-Series backbone. However, we consider whether we could achieve higher performance by leveraging instruction models that has undergone unimodal math post-training, and thus compare Qwen2.5-7B-Instruct[1] and Qwen2.5-Math-7B-Instruct[2]. After completing the VL alignment stage, we conduct a small-scale comparative experiment on MMathCoT-1M, fine-tuning on 50K examples. Finally, our results show that using Qwen2.5-Math-Instruct as the backbone yields an advantage of approximately 1 percentage point on MathVision and MathVerse. Therefore, we include Qwen2.5-Math-7B-Instruct as the LLM backbone for subsequent experiments.

# E    Ablation Studies

## E.1    Effectiveness of Different Data Category

In the first stage, we mainly synthesized large-scale multiclass CoT data.

- **w/o $\mathcal{S}_{Ao}$**: In this variant, the answer-only data is reverted to its original format. This directly mimics the training mode used by models such as Math-LLaVA and Math-PUMA, which involves hybrid training on both direct answers ('fast thinking') and CoT thinking.
- **w/o $\mathcal{S}_{An}$**: This data will be replaced with its original organizational structure, where the analysis and final answer are provided in a free-form text format.
- **w/o $\mathcal{S}_C$**: This batch of data will be replaced with reasoning expressed in mathematical formal language, better reflecting symbolic and 'plan and reasoning' forms of reasoning.

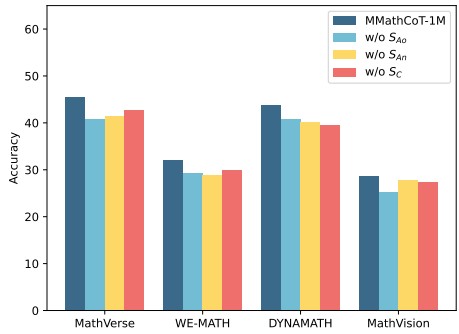

Figure 9: Each synthesis strategy towards different type of source data works well.

The results are shown in Figure 9. Firstly, it is shown across all datasets that using the complete synthesized data achieves the best results, highlighting the role of MMathCoT-1M data. More specifically, we find: i) $\mathcal{S}_{Ao}$ demonstrates the greatest impact on MathVerse and MathVision, indicating that expanded CoT data is important for problems where absolute solution accuracy is pursued; ii) However, on WE-MATH, the replacement of $\mathcal{S}_{An}$ leads to the most significant performance drop, suggesting that content rewriting better aligns with the end-to-end requirements posed by the WE-MATH benchmark, and mixing training with data lacking clear logical sequences may reduce hierarchical accuracy; iii) The results on DYNAMATH indicate that rewriting and natural language formulation effectively enhance reasoning robustness from the perspective of textual diversity. This reveals that the thought pattern in textual form tends to maintain the stability of the thought process more effectively under scenarios involving image transformations.

---

[1]https://huggingface.co/Qwen/Qwen2.5-7B-Instruct
[2]https://huggingface.co/Qwen/Qwen2.5-Math-7B-Instruct

## E.2 Selection of External Closed-source MLLM

In this section, we primarily present a comparison of metrics between Gemini-1.5-Flash-002[3] and other popular MLLMs, as well as a comparison on partial training data.

Table 12: SFT Performance with 50K data synthesized by two closed-source MLLMs, respectively.

| Model | MathVista-GPS | MathVerse | MathVision |
|---|---|---|---|
| URSA-8B w/ GPT-4o | 54.1 | 33.3 | 18.8 |
| URSA-8B w/ Gemini-1.5-Flash-002 | 55.1 | 32.5 | 18.3 |

- *Metrics Comparison*: We compare the performance of Gemini-1.5-Flash-002, GPT-4o, and GPT-4o-mini on some math-related tasks, as shown in Table 13. We observe that Gemini-1.5-Flash-002 is a MLLM that performs well on both unimodal and multimodal math tasks, and GPT-4o does not have a significant advantage over it. This, to some extent, ensures the quality of the data synthesis.
- *SFT Performance*: To best illustrate the performance variations, we randomly sampled 50K data sources from MMathCoT-1M, applied three corresponding strategies using GPT-4o, and subsequently conducted SFT. The performance results are shown in Table 12. We observe that using GPT-4o did not provide a clear advantage. However, the construction of MMathCoT-1M and DualMath-1.1M involves approximately 2.7 million API calls. The output token cost of Gemini-1.5-flash-002 is the same as that of GPT-4o-mini and is one-thirty-third of that of GPT-4o[4]. Therefore, Gemini-1.5-Flash-002 becomes a cost-effective choice.

However, we must say that if community researchers can afford the cost of accessing more powerful closed-source models, we expect the results to be even better.

Table 13: Comparison of Model Performance on Math Benchmarks

| Model | Avg | MATH | MathVista | MathVerse | MathVision |
|---|---|---|---|---|---|
| GPT-4o | 55.4 | 76.6 | 63.8 | 50.8 | 30.4 |
| Gemini-1.5-Flash-002 | 53.6 | 79.9 | 58.4 | 49.4 | 26.3 |
| GPT-4o-mini | 48.0 | 70.2 | 56.7 | 42.3 | 22.8 |

---

[3]https://deepmind.google/technologies/gemini/flash/
[4]https://docsbot.ai/models/gemini-1-5-flash-002

## F  Theoretical Analysis

Our theoretical explanation is grounded in the Q-value ranking theorem.

**Theoretical Background**   In a multi-step reasoning Markov Decision Process, the optimal Q-function, $Q^*(s_i, a_i)$, is the expected probability of reaching the correct final answer after action $a_i$. The scalar reward $r_{si}$ from our URSA-8B-RM at each step serves as an empirical estimate, $Q_{est}$, of this true Q-value.

**Theoretical Optimal Objective**   Approximating the ideal Q-value Ordering, under an optimal reasoning policy $\pi^*$, the Q-values along a reasoning path should satisfy a strict monotonic ordering: $Q^*_{w_{|W|}} < \cdots < Q^*_{w_1} \ll V^*(x) < \cdots < Q^*_{c_1} < Q^*_{c_{|C|}}$, where $c_i$ and $w_j$ denote the i-th correct step and the j-th incorrect step, respectively. In our drop-moment definition, $c_{|C|}$ is adjacent to $w_1$, which is consistent. The ideal training objective for a Process Reward Model (PRM) is to learn a Q-estimator $Q_{est}$ that strictly follows this ordering across all reasoning paths. For a successful reasoning path $\tau$ (e.g., $o(\tau) = 1$), this implies that the sequence of $Q_{est}$ outputs $\{r_{p,1}, r_{p,2}, \ldots, r_{p,T}\}$ should be monotonically increasing with no decreases, as a successful path should not contain any error.

Thus, the theoretically optimal generation objective is:

$$J_{ideal}(\theta) = \mathbb{E}_{\tau \sim \pi_\theta}[o(\tau)] - \lambda \cdot \Omega(G(\tau))$$

where:

- $\mathbb{E}_{\tau \sim \pi_\theta}[o(\tau)]$ is the standard final result accuracy.
- $\Omega(G(\tau))$ is a non-decreasing regularization term measuring the violation of the ideal Q-value ordering. $\lambda$ is its strength.
- $G(\tau) = r_{c_l}(\tau) - r_{w_1}(\tau)$ is a direct measure of the violation of the theoretical gap. The larger $G(\tau)$, the greater the drop from high (correct) to low (incorrect) rewards at some point in the path, which severely violates the ideal ordering. Minimizing $G(\tau)$ is equivalent to aligning the model's generation process with the Q-value dynamics of the optimal reasoning path.

**PS-GRPO as a Proxy Implementation**   PS-GRPO introduces a computable, differentiable proxy for $\Omega(G(\tau))$ into its reward function:

$$R(\tau) = o(\tau) \cdot (1 - \gamma \cdot \mathbb{I}(\delta_p(\tau) \geq \rho))$$

This function approximates the ideal objective $J_{ideal}(\theta)$:

- The $o(\tau)$ term directly corresponds to the first term in $J_{ideal}$, maximizing final accuracy.
- The penalty term approximates the regularizer:
    - $\delta_p(\tau)$ is a computationally tractable and strongly correlated proxy for $G(\tau)$. A large $\delta_p$ indicates a large $G$.
    - The indicator function $\mathbb{I}(\delta_p(\tau) \geq \rho)$ converts the continuous $\delta_p$ into a binary signal, resolving non-differentiability and ensuring compatibility with GRPO.
    - The hyperparameter $\rho$ acts as the violation threshold for $\Omega(G(\tau))$ and is determined by the behavior of $Q_{est}$.

Therefore, the PS-GRPO reward function $R(\tau)$ serves as a practical proxy for $J_{ideal}$: $J_{PS-GRPO}(\theta) \approx J_{ideal}(\theta)$. The final step is to empirically determine $\rho$ for a given PRM.

# G Implementation Details

## G.1 RL Data Curation

After instruction fine-tuning on MMathCoT-1M, the over-all accuracy did not exceed 50%. Therefore, we believe it still has the potential to be directly utilized in the RL phase. We collect 20K data with a types mixture ratio similar to that of instruction fine-tuning and conduct a one-time static filtering before RL. Specifically, we use URSA-8B to perform 8 samplings on this 20K data, filtering out examples where all 8 sampling results are either incorrect or correct. This left approximately 15K+ data for training vanilla GRPO and PS-GRPO. We implement PS-GRPO using TRL [104, 106]. The statistics of the RL data can be found in the table 10.

Figure 10: Statistics of RL data for vanilla GRPO and PS-GRPO.

| Statistic | Number |
|---|---|
| *Total Prompts* | |
| - Total number | 15.3K |
| *Data Source* | |
| - MathV360K | 2.7K |
| - Multimath-EN | 7.5K |
| - MAVIS-Geo | 2.2K |
| - MAVIS-MetaGen | 0.9K |
| - Geo170K-QA | 2.1K |
| *Problem Category Statistics* | |
| - Plane Geometry | 4.1K (26.8%) |
| - Analytic Geometry | 1.9K (12.4%) |
| - Solid Geometry | 1.0K (6.5%) |
| - Algebra | 1.1K (7.2%) |
| - Function | 2.9K (19.0%) |
| - Chart | 1.9K (12.4%) |
| - Textbook QA | 0.9K (5.9%) |
| - Formula | 0.8K (5.2%) |
| - Arithmetic | 0.7K (4.6%) |

## G.2 Parameters and Time Cost

In this section, we provide the specific parameter settings and time costs for the three stages. Our experiments are based on Python 3.10 and PyTorch 2.4.0+cu124. We use AdamW [107] as the optimizer. We use Fully Shared Data Parallel (FSDP) [108] as the distributed training framework. Unless otherwise specified, experiments are conducted on 32× NVIDIA-H100-HBM3 GPUs by default. Additionally, we provide important parameters used in data construction. During the generation of positive and negative example pairs, we set the $temperature$ to 1.0, $n\_return\_sequences$ to 16, and $top\_p$ to 0.95. In the *BinaryErrorLocating* phase, we set the $temperature$ to 0.3, $n\_return\_sequences$ to 16, and $top\_p$ to 0.95.

We adapt the vLLM [109] framework for the URSA-8B's architecture (hybrid vision tower + MLP + Qwen2.5-math-Instruct is not originally supported by VLLM) and use it as an acceleration tool during the inference phase. During the data pair generation phase, we use 16× NVIDIA-H100-HBM3 GPUs for inference, which takes approximately 28 hours. In the *BinaryErrorLocating* phase, we also use 16× NVIDIA-H100-HBM3 GPUs for inference, taking about 20 hours.

The hyperparameter and time cost used in Stage I and Stage II are demonstrated in Table 14. Since the parameters used in Stage III are somewhat different, we list them separately in Table 15. Recently, much work has provided numerous optimization tricks for GRPO, such as training-time dynamic sampling, clipping higher values, abandoning KL loss, etc [110, 58]. However, to independently verify the effectiveness of PRM-guided reward modeling, we have not added these tricks in either vanilla GRPO or PS-GRPO to ensure a fair and valid verification process. We only do **one-time** difficulty-based data selection before applying RL.

Table 14: Hyperparameter setting and training time cost in Stage I and II.

| Hyperparameters & Cost | VL-alignment | Instruction Fine-tuning | PRM Training |
|---|---|---|---|
| Learning Rate | 1e-4 | 1e-5 | 5e-6 |
| Epoch | 1 | 2 | 2 |
| Warm-up Ratio | 0.02 | 0.02 | 0.02 |
| Weight Decay | 0.02 | 0.01 | 0.02 |
| Batch Size | 64 | 128 | 128 |
| Trainable Parts | Aligner | Vision Encoder, Aligner, Base LLM | Base LLM |
| Data Size | 860K | 1.0M | 1.1M |
| Time Cost | ∼3.5h | ∼11h | ∼12h |

## G.3 Benchmarks

In this section, we introduce the detailed subtasks and metrics of four used benchmarks to more precisely demonstrate the evaluation.

**MathVerse**  MathVerse [74] is a benchmark for testing the reasoning abilities of MLLMs when the information content in text and image modalities varies. Specifically, the models focus on performance in six scenarios: Text-Dominant (TD), Text-Lite (TL), Text-Only (TO), Vision-Intensive (VI), Vision-Dominant (VD) and Vision-Only (VO).

**WE-MATH**  WE-MATH [77] is the first benchmark that decompose composite problems into sub-problems according to the required knowledge concepts. In figure 9, the actual content corresponding to the abbreviations is as follows. Mem: Measurement, PF: Plane Figures, SF: Solid Figures, TMF: Transformations and Motion of Figures, PD: Position and Direction, AL: Angles and Length, UCU: Understanding and Conversion of Units, CPF: Calculation of Plane Figures, UPF: Understanding of Plane Figures, CSF: Calculation of Solid Figures, USF: Understanding of Solid Figures, BTF: Basic Transformations of Figures, CCF: Cutting and Combining of Figures, Dir: Direction, Pos: Position, RoM: Route Map, CCP: Correspondence of Coordinates and Positions.

Table 15: Hyperparameter setting and training time cost in Stage III.

| Hyperparameters / Cost | Value |
|---|---|
| Epochs | 2 |
| Learning Rate | 2e-6 |
| Temperature | 1.0 |
| Rollout number per prompt | 8 |
| Prompt Max Length | 6048 |
| Output Max Length | 3072 |
| Precision | bf16 |
| Train Batch Size | 512 |
| KL Coefficient | 0.003 |
| Data size | 15K |
| Time Cost | ~18h |

**DYNAMATH**  DYNAMATH [75] is a benchmark designed to evaluate the robustness of MLLMs in mathematical reasoning. Specifically, it includes tests across multiple dimensions, including Solid Geometry (SG), Plane Geometry (PG), Analytic Geometry (AG), Algebra (AL), Puzzle Test (PT), Graph Theory (GT), Arithmetic (AR), Scientific Figure (SF) and Statistics (ST). It includes 501 seed questions and 5010 generated questions.

**GeoQA**  The GeoQA [40] dataset is a specialized dataset designed for evaluating and training models in the field of geographic question answering. Its test set includes 734 samples.

**MathVista**  MathVista [76] comprises a total of 5 subtasks: Geometry Problem Solving (GPS), Math Word Problem (MWP), Figure Question Answering (FQA), Textbook Question Answering (TQA) and Visual Question Answering (VQA). Like the previous math reasoning works, our model training process does not overly focus on knowledge-intensive tasks (such as VQA and FQA), hence we choose GPS as the primary task.

**MathVision**  MathVision [111] is a large-scale multimodal math reasoning dataset that broadens the disciplinary scope of the multimodal mathematics field. The test set contains 3,040 examples, covering 16 key competencies, and provides reliable testing performance. Specifically, The specific meanings of the various disciplinary indicators in Table 6 are listed as following. Alg: algebra, AnaG: analytic geometry, Ari: arithmetic, CombG: combinatorial geometry, Comb: combinatorics, Cnt: counting, DescG: descriptive geometry, GrphT: graph theory, Log: logic, Angle: metric geometry - angle, Area: metric geometry - area, Len: metric geometry-length, SolG: solid geometry, Stat: statistics, Topo: topology, TransG: transformation geometry.

**Evaluation Criteria**  Our comparison is based on the following criteria: First, we select the results from the official leaderboards of each benchmark. Second, we choose the results from the original papers or technical reports of each model. Finally, we conduct our own inference and evaluation using vLLM [109]. Our evaluation adheres to the rules of the benchmarks themselves, which are as follows:

- **Rule-based Matching**: WEMATH, GeoQA.
- **LLM-as-a-Judge**: MathVista, MathVision, MathVerse, Dynamath.

The prompt for LLM-as-a-Judge is shown in Figure 11.

### G.4  Algorithm

In this section, we place the specific process of BIE from Section 3.2 into Algorithm 1. Specifically, the input is a solution that points to an incorrect answer. We set a per-step sampling hyperparameter

Below are two answers to a math question. Question is [Question], [Standard Answer] is the standard answer to the question, and [Model_answer] is the answer extracted from a model's output to this question.

Determine whether these two answers are consistent.

Please note that only when the [Model_answer] completely matches the [Standard Answer] means they are consistent. For non-multiple-choice questions, if the meaning is expressed in the same way, it is also considered consistent, for example, 0.5m and 50cm.

If they are consistent, Judement is 1; if they are different, Judement is 0.

## Example 1:
[Question]: Write the set of numbers represented on the number line in interval notation.
[Standard Answer]: (-2,1]
[Model_answer] : Extracted Answer: \((-2, 1)\)
**Judgement:** 0

## Example 3:
[Question]: Find the domain and range of the function f using interval notation.
[Standard Answer]: domain: [-4, 0) and range: (-3, 1]
[Model_answer] : Range: \((-4, 1]\)
**Judgement:** 0

OK. Now let's begin:
[Question]: {QuestionText}
[Standard Answer]: {AnswerText}
[Model answer]: {ResponseText}

Figure 11: LLM-as-a-Judge prompt used for answer matching.

$N_{\text{mid}}$. Initially, we set the start and end points of the search range to Step 1 and Step N, respectively. We first consider the $mc$ value of Step (1+N)//2. If it is positive, it indicates that the first totally erroneous step occurs in the latter half; otherwise, we look in the first half. This reduces the number of searches to $\mathcal{O}(\log N)$.

Besides, we introduce the process of PS-GRPO in Algorithm 2. This process involves merging of outcome reward and process-as-outcome reward, and subsequent relative advantages calculation.

**Algorithm 1** BinaryErrorLocating

1: **Input:** Solution set $S = \{y_i | i = 1, 2, ..., N\}$, mid step sampling num $N_{mid}$.
2: **for** $i = 1$ **to** $N$ **do**
3:     $l \leftarrow 0, r \leftarrow$ StepLen$(y_i)$                              ▷ Define start and end index.
4:     **while** $l < r$ **do**
5:         $mid \leftarrow \lfloor (l+r)/2 \rfloor$
6:         MID_RES $\leftarrow \mathcal{F}(\{y_{i,1}, y_{i,2}, ..., y_{i,mid}\}, N_{mid})$        ▷ Sampling from mid step.
7:         **if** VERIFY(MID_RES) does not contain True **then**    ▷ Judge sampling's final answer.
8:             $r \leftarrow mid$
9:         **else**
10:             $l \leftarrow mid + 1$
11:         **end if**
12:     **end while**
13:     $e_i \leftarrow$ WRONGSTEPLABELING$(y_i, l)$                 ▷ Collect sequential labels of $y_i$.
14:     ErrorLocatingSet.append$\{e_i\}$
15: **end for**
16: **Output:** ErrorLocatingSet

---

**Algorithm 2** PS-GRPO

1: **Input:** Policy model $\pi_\theta$, Process Reward Model $\phi_\theta$, train dataset $D_{RL} = \{I_i, Q_i, Y_i\}_{i=1}^{N}$.
2: ErrorLocatingSet $\leftarrow$ []
3: **for** Epoch $= 1$ $to$ $N$ **do**
4:     **for** Batched data $\{I, Q, Y\}$ $in$ $D_{RL}$ **do**
5:         Generate rollouts of $\{I, Q\}$: $\{c^j\}_{j=1}^{M} \sim \pi_\theta$
6:         Calculate process reward sets $S = \{(p_1, p_2, \cdots, p_{|c_j|}) \sim \phi_\theta(c^j) | 0 < j < M\}$
7:         Calculate reward $\{r^j\}_{i=1}^{M}$ by Equation 5 and 6.
8:         Calculate relative advantages $\{\hat{A}^j\}_{i=1}^{M}$ using Equation 7.
9:         Update policy model $\pi_\theta$ using Equation 8.
10:     **end for**
11: **end for**
12: **Output:** Updated policy model $\pi_\theta$.

# H   Prompt Design

## H.1   Prompt Utilized in MMathCoT-1M Synthesis

In this section, we provide the specific prompts for three-module data synthesis. Additionally, Gemini-1.5-Flash is a model that is very sensitive to prompts and parameters in practical experience, and we will share detailed adjustment experiences.

**CoT Expansion** CoT expansion prompt for answer-only data source can be seen in Figure 12. We order the Gemini-1.5-flash to give a reasonable process directing to the ground-truth. After the execution, we find that the outputs is not so clear. The model sometimes will give trajectories that include "we must trust the answer" or "let me assume". We identify these phrases as signals that the model can not solve the problem naturally and independently. We will filter these samples.

**Analysis Rewriting** Rewriting prompt for analysis-formatted data synthesis is illustrated in Figure 13. For solutions in an analytical format, we transform them into clear step-by-step format trajectories. During this process, Gemini-1.5-flash-002 does not exhibit significant questioning or make conditional requests. We improve data quality through reorganization and polishing of the language logic.

**Format Unified** By employing a unified format prompt shown in Figure 14 to modify the reasoning styles of plan-and-reasoning and symbolic approaches, we are able to extract a more natural language process aligned with the pre-training style. A single example is sufficient to elicit perceptually favorable responses.

Figure 12: CoT expansion prompt for answer-only data.

**Double-checking**  After completing the above three points, we apply an LLM-as-a-judge for double-checking the synthesized data, ensuring that the solutions do not contain unreasonable processes, such as untimely questioning, conditional requests, or reasoning loops. The specific prompt design is shown in Figure 15. After this layer of filtering, we obtain the final MMathCoT-1M.

### H.2  Prompt Utilized in DualMath-1.1M Synthesis

In this section, we demonstrate the prompts used in MIE.

- **Geometry Problem**: For geometry problem, we prompt the Gemini-1.5-Flash-002 to first identify key geometry features in the figure. We then order it to introduce a misinterpretation on these elements. Finally, use the wrong information to execute a misleading solution. The total design can be seen in Figure 16.

I have a mathematical problem-solving process here, and both the process and the final answer are completely correct. I hope you can solve it in a **different way** based on the existing solution process, ensuring that the process and results are correct, especially the reasoning results **remain consistent with the original**.

1. You must **trust** that the solution and final answer is correct, and you **do not need to review or refute it**.

2. Your transcription must be **semantically coherent**.

3. You **cannot make requests** like "require more information" or similar, because the given conditions are sufficient and the solution is definitely correct.

4. You can rewrite from multiple perspectives, such as **providing different solutions** to the problem or adopting **different textual styles**.

5. You must end with '†Answer: ', filling in the definitive answer provided.

Question:
{QuestionText}
Correct Solution:
{SolutionText}
Give the rewrited solution in the following format:
Step 1: ...
Step 2: ...
...
†Answer: ...

Figure 13: Analysis rewriting prompt for analysis-formatted data.

- **Charts & Function**: For ChartQA and math functions, we prompt Gemini-1.5-Flash-002 to first check the fine-grained data points. We then attempt to insert spatially similar data to induce a misinterpretation. This subsequently leads to incorrect solutions for automatic labeling.
- **LLM-as-a-Judge**: For chart reasoning and function problem, we execute similar process on Gemini-1.5-Flash-002. We place it in Figure 17.

# I   Case Study

## I.1   Showcase on Best-of-N Evaluation

To more clearly illustrate the effectiveness of URSA-8B-RM in BoN evaluation, a case on MathVista-GPS is demonstrated (Figure 18). This case shows that URSA-8B-RM is sensitive to false theorem application and misunderstandings of angle-number relations. The good property not only enables URSA-8B-RM to perform well in BoN evaluation but also endows it with the potential to identify more valuable learning samples in online reinforcement learning.

## I.2   Process of Misinterpretation Insertion Engine

As shown in Figure 19, MIE performs three main actions: First, it interprets the mathematical information in the image. Then, it replaces key information at a selected step. Finally, it continues reasoning based on the modified conditions.

## I.3   Failure Pattern in Process During GRPO

In this section, we intuitively reveal through examples why PS-GRPO works effectively. We first introduce the concept of false-positive rollouts, which are rollouts that, despite reaching the correct answer, do not provide perfect intermediate actions to arrive at the solution. They can generally be

You are given a question and a solution, and your task is to translate the reasoning process from mathematical language into a style of natural language reasoning.

Instructions:

- Convert the formal mathematical solution into a **natural language explanation**.

- Use "Step X:" to separate each step.

- End the solution with "†Answer: [final answer]".

- Ensure the explanation is clear, concise, and free of formal mathematical symbols unless absolutely necessary.

- **You must strictly follow the provided solution steps and final answer. Do not modify, correct, or reinterpret any part of the solution.**

## Example:

Question:

Given that OB is perpendicular to OA, line CD passes through point O, and ∠AOC = 20°, then ∠BOD = ?

Formal Solution:

Step 1 (Definition of Perpendicular): ∵ OB ⊥ OA, ∴ ∠BOA = 90°.

Step 2 (Given condition): ∵ ∠AOC=20°, ∴ ∠BOC = 90° - ∠AOC = 90° - 20° = 70°.

Step 3 (Supplementary angles property): ∠BOD = 180° - ∠BOC = 180° - 70° = 110°.

Answer: \\boxed{110}

Your Response:

Step 1: Since OB is perpendicular to OA, we have that angle BOA is 90 degrees.

Step 2: Given that angle AOC is 20 degrees, it follows that angle BOC is 90 degrees minus angle AOC, which is 70 degrees.

Step 3: Using the property of supplementary angles, angle BOD is 180 degrees minus angle BOC, which is 110 degrees.

†Answer: 110

Now, let's begin!

Question:

{QuestionText}

Solution:

{SolutionText}

** Your Response:

Figure 14: Format unify prompt for mathematical and symbolic reasoning style data.

divided into two categories: (i) the lack of visual condition alignment. Solutions in this category exhibit inconsistencies in reasoning regarding basic visual factors such as edge relationships, coordinate values, and theorem applications, revealing deficiencies in the pretraining phase, as shown in Figure 20. (ii) the exploitation of shortcut patterns. These rollouts do not go through key steps but are guided directly to the correct answer after basic descriptions due to the high correlation between image features and problem-solving patterns during pretraining or SFT, as shown in Figure 21. Therefore, PS-GRPO suppresses the advantageous direction brought by these erroneous actions through the sensitivity of the PRM in online RL for error identification. This leads to a more optimal paradigm that combines outcome rewards with process reward-based penalties.

## I.4 Cases on How Naive Process Reward Modeling Fails

In this section, we elaborate on the two fundamental flaws of process reward guided RL mentioned in Section 4 and present some cases for illustration. In online RL, models can easily recognize the patterns for obtaining process rewards, leading to conservative analyses and concise responses as they sidestep PRM scrutiny. As shown in Figure 22, we have observed that the model often follows a distinct reasoning pattern. They initially read and analyze the given conditions comprehensively, but then make incorrect decisions based on this analysis, leading to wrong answers. This indicates that

> **Double Checking**
>
> You are tasked with evaluating whether a provided solution **aligns precisely** with a given standard answer. Your evaluation must strictly adhere to the following criteria:
>
> ### Solution Fidelity
>
> - **Certainty**: The solution must be free of any elements of doubt or speculative reasoning. Avoid using phrases that imply uncertainty or hypothetical reasoning, such as "more conditions are needed", "the answer seems questionable", "let us assume", "the provided solution seems wrong" or any similar expressions. The solution should be presented with confidence and clarity, reflecting a definitive and well-supported conclusion.
>
> ### Solution Consistency
>
> - **Alignment with Standard Answer**: The final conclusion of the solution must match the given standard answer exactly. The reasoning process should directly and unequivocally lead to the provided answer without deviation. Ensure that there are no discrepancies between the solution's conclusion and the standard answer.
>
> ### Evaluation Process
>
> 1. **Analyze the Solution**: Carefully review the logical steps and reasoning used in the solution to ensure they are sound and lead directly to the conclusion.
>
> 2. **Compare with Standard Answer**: Verify that the conclusion of the solution is **identical to the standard answer** provided.
>
> 3. **Provide Judgment**: Based on your analysis, offer a clear and concise judgment
>
> finally:
>
> - Respond with "**Judgment: yes**" if the solution's conclusion is consistent with the given standard answer.
>
> - Respond with "**Judgment: no**" if the solution's conclusion does not match the standard answer.
>
> ### Example Structure
> - **Question**:
> {QuestionText}
> - **Standard Answer**:
> {AnswerText}
> - **Solution Answer**:
> {SolutionText}
> ### Your Response

Figure 15: Double-checking prompt for ensuring high-quality and appropriate trajectories in synthesized CoT reasoning data.

when explicitly modeling process rewards, models can easily focus on processes that seem "correct" in isolation. However, these processes may not be genuinely helpful for the final outcome and instead may lead the model to prioritize high process rewards over accuracy.

## J Limitations

Our work requires certain computational power to be efficient. We share high-quality training data in the community through open-sourcing, though it comes at a cost. Our exploration mainly focuses on STEM tasks, and it can be expected to work on more general visual domains. Our work primarily focuses on enhancing STEM reasoning capabilities and does not raise fairness or ethics issues.

## K Broader Impact

Our work primarily focuses on the STEM reasoning capabilities of MLLMs. It is highly likely to have a positive impact on society, as this direction can make MLLMs more reliable tools. A potential negative impact could be that with the advancement of this technology, certain groups may become dependent on this capability.

**Misinterpretation Insertion for Geometry Problem**

You are given a geometry problem with an image and its solution. Your task is to introduce errors into the solution by misreading the diagram and generate an incorrect answer.
Instructions:

- Your response involves three Stages:
  ** Stage 1: Analyze the Correct Solution **: Identify where in the solution diagram information is extracted. Note these areas as **potential points for introducing misinterpretations**.
  ** Stage 2: Introduce a Misinterpretation **: Choose one of the identified points from Stage 1 and **alter it to create a misleading scenario**. Integrate this misinterpretation naturally into the solution.
  ** Stage 3: Continue Reasoning **: Based on the misinterpretation from Stage 2, **continue the reasoning** process to derive an incorrect final answer.

- Ensure the misreading is **naturally integrated** without explicit statements about making a misinterpretation. When describing corrupted solution, avoid showing the misleading thought process. You must respond in a typical problem-solving style.
- Remember that the <pos> and <neg> tags only need to appear at the end of each Step X. Do not repeat them between Step X and Step X+1, even if the reasoning process for that step spans many lines.
- Remember that once a step is marked as <neg>, all subsequent steps are also considered <neg>.
- '†Answer:' is the final answer, and it should not be tagged.
- You **cannot** tag an unaltered step with <neg> without misinterpreting it. You must misinterpret the step if you are tagging it with <neg>.
- Words that imply misinterpretation, such as **'misread' 'incorrect' or 'incorrectly assume'** must not appear in your response under any circumstances.
- The misinterpretation action in Stage 3 should be **consistent** with what was planned in Stage 2.

Question:
{QuestionText}

Solution:
{SolutionText}

** Your response:

Figure 16: Misinterpretation insertion for geometry-related problems.

**Misinterpretation Insertion for Charts & Function**

You are given a math problem with a **coordinate axis or chart** and its solution. Your task is to introduce errors into the solution by misreading the image or chart and generate an incorrect solution.
Instructions:

- Your response involves three Stages:
  ** Stage 1: Image Information Extraction **: Identify where in the solution diagram information is extracted. Note these areas as potential points for introducing misinterpretations.
  ** Stage 2: Introduce a Misinterpretation **: Choose one of the identified points from Stage 1 and alter it to create a misreading scenario. Integrate this misinterpretation naturally into the solution.
  ** Stage 3: Continue Reasoning **: Based on the misinterpretation from Stage 2, continue the reasoning process to derive an incorrect final answer.

- Ensure the misreading is naturally integrated without explicit statements about making a misinterpretation. When describing corrupted solution, avoid showing the misleading thought process. You must respond in a typical problem-solving style.
- Remember that the <pos> and <neg> tags only need to appear at the end of each Step X. Do not repeat them between Step X and Step X+1, even if the reasoning process for that step spans many lines. Remember that once a step is marked as <neg>, all subsequent steps are also considered <neg>.
- '†Answer:' is the final answer, and it should not be tagged.
- You cannot tag an unaltered step with <neg> without misinterpreting it. You must misinterpret the step if you are tagging it with <neg>.
- Words that imply misinterpretation, such as 'misread' 'incorrect' or 'incorrectly assume' must not appear in your response under any circumstances.
- The misinterpretation action in Stage 3 should be consistent with what was planned in Stage 2.

Question:
{QuestionText}

Solution:
{SolutionText}

** Your response:

Figure 17: Misinterpretation insertion for function and chart-related problems.

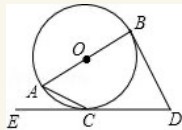

**Problem**:

As shown in the figure, AB is the diameter of circle O, DB and DC are respectively tangent to circle O at points B and C. If ∠ACE = 25.0°, then the degree of ∠D is ()

A. 50° B. 55° C. 60° D. 65°

**Solution 1**:

Step 1: We are given that AB is the diameter of circle O, and DB and DC are tangents to circle O at points B and C respectively. We are also given that ∠ACE = 25°. 0.93

Step 2: Since DB and DC are tangents to circle O from the same point D, we know that DB = DC. This means that triangle DBC is an isosceles triangle. 0.95

Step 3: In triangle DBC, let ∠DBC = ∠DCB = x. Since the sum of angles in a triangle is 180°, we have ∠BDC + ∠DBC + ∠DCB = 180°. Therefore, ∠BDC + 2x = 180°. 0.91

Step 4: Since AB is the diameter, ∠ACB is a right angle (90°). In right-angled triangle ACB, we have ∠CAB + ∠ABC = 90°. 0.93

Step 5: We are given that ∠ACE = 25°. Since ∠ACB = 90°, we have ∠ECB = 90° - 25° = 65°. 0.98

Step 6: In triangle DBC, ∠DBC + ∠DCB + ∠BDC = 180°. Since DB and DC are tangents, ∠DBC = ∠DCB = 90° - 25° = 65°. Therefore, 65° + 65° + ∠BDC = 180°, which gives ∠BDC = 180° - 130° = 50°. 0.98

Step 7: Therefore, ∠D = ∠BDC = 50°. 0.88

†Answer: A ✅

**Solution 2**:

Step 1: Identify the relevant angles. We are given that ∠ACE = 25°. We want to find ∠D. 0.99

Step 2: Recognize the relationship between the angles. Since AB is the diameter of circle O, and DB is tangent to circle O at B, we know that ∠ABD is a right angle (90°). Also, since DB and DC are tangents from the point D to circle O, we know that DB = DC. This means that triangle DBC is an isosceles triangle. 0.94

Step 3: Use the property of isosceles triangles. In isosceles triangle DBC, ∠DBC = ∠DCB. Let's denote this angle as x. The sum of angles in triangle DBC is 180°. Therefore, ∠BDC + ∠DBC + ∠DCB = 180°. Substituting the values, we get ∠BDC + x + x = 180°, which simplifies to ∠BDC + 2x = 180°. 0.90

Step 4: Relate the angles. ∠ACE is an inscribed angle subtending the arc BC. The angle subtended by the same arc at the center is twice the inscribed angle. Therefore, ∠BOC = 2 * ∠ACE = 2 * 25° = 50°. 0.46

Step 5: Find the angle ∠BDC. Since ∠ABD is a right angle (90°), and ∠BOC is 50°, we can find ∠BDC. In quadrilateral ABCD, the sum of angles is 360°. We have ∠DAB + ∠ABC + ∠BCD + ∠CDA = 360°. Since ∠ABC = 90° and ∠BCD = x, we have ∠DAB + 90° + x + ∠CDA = 360°. 0.67

Step 6: Solve for x. We know that ∠BDC + 2x = 180°. Also, ∠BDC + ∠DAB = 180° (angles on a straight line). Therefore, ∠DAB = 180° - ∠BDC. Substituting this into the equation from step 5, we get (180° - ∠BDC) + 90° + x + ∠CDA = 360°. This simplifies to 270° - ∠BDC + x + ∠CDA = 360°. 0.65

Step 7: Determine ∠D. Since triangle DBC is isosceles, ∠DBC = ∠DCB = x. Also, ∠BDC + 2x = 180°. We know that ∠ACE = 25°, which is an inscribed angle subtending arc BC. The central angle subtending the same arc is 2 * 25° = 50°. This central angle is ∠BOC. In triangle OBC, OB = OC (radii), so it's an isosceles triangle. Therefore, ∠OBC = ∠OCB = (180° - 50°)/2 = 65°. Since ∠OBC and ∠DBC are supplementary, ∠DBC = 180° - 65° = 115°. However, this is incorrect. Let's reconsider. Since ∠ACE = 25°, the arc BC subtends an angle of 50° at the center. In triangle OBC, OB = OC, so ∠OBC = ∠OCB = (180° - 50°)/2 = 65°. Therefore, ∠D = 65°. 0.47

†Answer: D ❌

Figure 18: Case of URSA-8B-RM serving as a verifier in Best-of-N evaluation.

**Showcase of how MIE works**

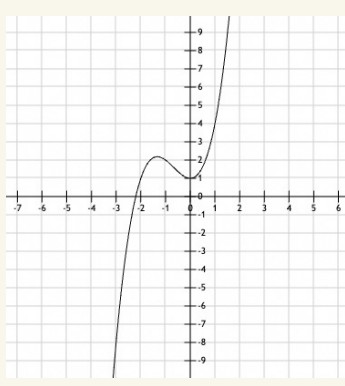

**Problem**:

Given a cubic function y=ax^3+bx^2+cx+d, please determine the values of a, b, c and d.

**Original Solution**:

Step 1: I first need to identify the points that the graph passes through, and then substitute these points into the equation to solve for the values of a, b, c, and d.

Step2: I find some points below: P1: (-3, -8) P2: (0, 1) P3: (-2, 1) P4: (-1, 2)

…

Step N: a=1, b=2, c=0, d=1

**Gemini's response:**

## Information Extraction: I see this question is highly related to the coordinates of points on the line. The line appears in the third quadrant, second quadrant, and first quadrant.

## Misinterpretation Insertion: A simple method is to introduce incorrect point coordinates to reach a wrong conclusion. As the correct solution states, the coordinates of point P1 are (-3, -8), but we easily replace them with (-3, -7) as a premise.

## Continue Reasoning:

Step 1: I first need to identify the points that the graph passes through, and then substitute these points into the equation to solve for the values of a, b, c, and d. \<pos\>

Step 2: I find some points below: P1: (-3, -7) P2: (0, 1) P3: (-2, 1) P4: (-1, 2). \<neg\>

Step 3: Using P1: (-3, -7),

$$-7=a(-3)^3+b(-3)^2+c(-3)+1$$
$$-7=-27a+9b-3c+1$$
$$-8=-27a+9b-3c \text{ \<neg\>}$$

…

Step N: $a = \frac{5}{6}, b = \frac{3}{2}, c = -\frac{1}{3}, d = 1.$ \<neg\>

Figure 19: Case from MIE. We introduce specific step-level perception errors and continue false reasoning to construct a correctnesses-labeled solution.

**Rollouts Analysis on Process (I)**

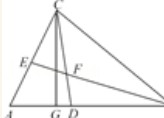

**Problem**:
In the diagram, point F is the intersection of the angle bisectors CD and BE of triangle ABC, and CG is perpendicular to AB at point G. If angle ACG is 32 degrees, what is the measure of angle BFC?

**Perfect Rollout:**

Step 1: Recognize that CD and BE are angle bisectors, so point F is the incenter of triangle ABC.

Step 2: Since CG is perpendicular to AB, triangle CGA is a right triangle.

Step 3: In triangle CGA, using the fact that the sum of angles is 180°, calculate angle CAG: angle CAG = 90° - angle ACG =90° - 32° = 58°.

Step 4: Since CD is the angle bisector of angle ACB, angle ACD = angle BCD = 1/2 angle ACB.

Step 5: Let angle ACB = 2θ. Then angle ACD = angle BCD = θ.

Step 6: Similarly, BE is the angle bisector of angle ABC, so angle ABE = angle EBC = 1/2 angle ABC.

Step 7: Let angle ABC = 2φ. Then angle ABE = angle EBC = φ.

Step 8: In triangle ABC, the sum of angles is 180°. So, angle BAC + angle ABC + angle ACB = 180°. Substituting angle BAC = 58°, and angle ABC = 2φ, angle ACB = 2θ. So, 58° + 2φ + 2θ = 180°, which simplifies to φ + θ = 61°.

Step 9: In triangle BFC, the sum of angles is 180°. So, angle BFC + angle FBC + angle FCB = 180°.

Step 10: Substitute angle FBC = φ and angle FCB = θ. So, angle BFC + φ + θ = 180°. From step 8, φ + θ = 61°, so angle BFC = 180° - 61° = 119°。

†Answer: 119 ° 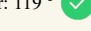

**False-positive Rollout:**

Step 1: Recognize that CD and BE are angle bisectors, so point F is the incenter of triangle ABC.

Step 2: Since CG is perpendicular to AB, triangle CGA is a right triangle.

Step 3: In triangle CGA, using the fact that the sum of angles is 180°, calculate angle CAG: angle CAG = 90° - angle ACG =90° - 32° = 58°.

Step 4: Angle BFC = 180° - angle CDE. CD is the angle bisector of angle ACB, angle ACD = angle BCD = 1/2 angle ACB.

Step 5: We have angle CDE = angle ACD = angle BCD = 1/2 angle ACB.

Step 6: Since angle ACB + angle ACG = 180°, angle ACB = 180° - 58° = 122°.

Step 7: Substitute angle ACB = 122°, angle ACD = 1/2 * 122° = 61°.

Step 8: Substitute angle ACD = 66°, angle BFC = 180° - 61° = 119°.

†Answer: 119 ° 

**Analysis:**

In this example, false-positive rollout leads to numerous instances of image-text perception inconsistency, and such behavior should not be encouraged in reasoning, even if it happens to yield the correct result.

Figure 20: False positive rollout analysis I.

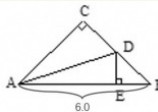

**Problem**:
As shown in the figure, AC = BC, AD bisects angle CAB, then the perimeter of triangle DBE is () Choices: A:6cm B:7cm C:8cm D:9cm

**Perfect Rollout:**

Step 1: The angle bisector states that a point on an angle bisector is equidistant from the two sides of the angle. Since AD bisects angle CAB, DC ⊥ AC, DE ⊥ AB, we conclude that DC=DE.

Step 2: Since angle C = angle AED = 90°, angle CAD = angle EAD, we have triangle ACD ≅ triangle AED.

Step 3: We substitute AC with BC, so BC = AE.

Step 4: The perimeter of triangle DBE equals to DB + BE + DE, notice that AE + BE = AB

Step 5: From the diagram. AB = 6cm

Step 6: The answer is 6cm.

†Answer: A 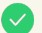

**False-positive Rollout:**

Step 1: Look at the figure and notice that AC = BC, so triangle ABC must be isosceles.

Step 2: AD is the angle bisector of angle CAB. Hmm, so it splits angle A into two equal parts.

Step 3: DE is perpendicular to AB.

Step 4: The question asks for the perimeter of triangle DBE. I need to find the lengths of DB, BE, and DE.

Step 5: Since ABC is isosceles with AC = BC, angles at A and B are equal.

Step 6: Maybe triangle DBE is related to triangle ABC in some way.

Step 7: Let me guess that the perimeter of DBE is equal to AB. If AB is 6 cm, then the perimeter of DBE would be 6 cm. So the answer is A.

†Answer: A 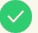

Analysis:
This type of learning pattern makes problems very difficult to solve when shortcuts in numerical and geometric relationships point in the wrong direction.

Figure 21: False positive rollout analysis II.

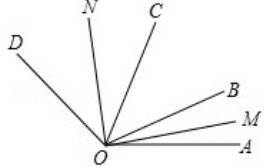

**Question:**
As shown in the figure: ∠AOB : ∠BOC : ∠COD = 2 : 3 : 4, ray OM and ON bisect ∠AOB and ∠COD respectively, and ∠MON = 84°, then ∠AOB is ( ).

**Solution:**
**Step 1**: Look at the diagram and note the angles ∠AOB, ∠BOC, and ∠COD are in the ratio 2:3:4.
**Step 2**: Recognize OM and ON bisect ∠AOB and ∠COD, creating smaller angles.
**Step 3**: Use the given ∠MON = 84° to set up an equation based on the ratios and angle bisectors.
**Step 4**: Solve the equation ∠AOB = 2 / (2 + 3 + 4) * 84° = 18.66°.
†Answer: ∠AOB is 18.66°.

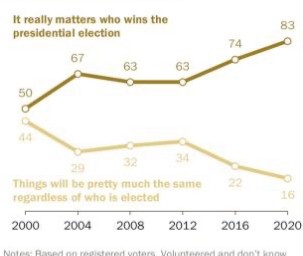

**Question:**
As shown in the figure: ∠AOB : ∠BOC : ∠COD = 2 : 3 : 4, ray OM and ON bisect ∠AOB and ∠COD respectively, and ∠MON = 84°, then ∠AOB is ( ).

**Solution:**
**Step 1**: The dark brown line represents voters who think it really matters, showing an increasing trend from 50% in 2000 to 83% in 2020. The light brown line represents voters who think things will be much the same, showing a decreasing trend from 44% in 2000 to 16% in 2020.
**Step 2**: We need to know the least value of the light brown graph to the leftmost value of the dark brown graph.
**Step 3**: The ratio of the least value of the light brown graph and the leftmost value of the dark brown graph is 16:67.
†Answer: 16:67.

Figure 22: Bad case analysis on two process reward modeling variants.

