# OpenReview forum: "Unlocking Multimodal Mathematical Reasoning via Process Reward Model"
_NeurIPS.cc/2025/Conference — NeurIPS 2025 poster_

### Official Review · Reviewer_GTaC · 2025-06-08

**Clarity:** 1
**Significance:** 2
**Originality:** 2
**Rating:** 3
**Confidence:** 3

**Summary:**

The paper presents URSA, a three-stage framework to enhance multimodal mathematical reasoning using PRMs. It addresses challenges like data scarcity and reward hacking. URSA constructs two large-scale datasets and introduces a PRM-aided online RL method, achieving superior performance over leading models like GPT-4o on multiple benchmarks.

**Questions:**

1. The notation in the PS-GRPO equations is confusing: Equation (5) uses subscript *p* for process reward, while Equation (9) in the appendix uses *s*; in Equation (6), it's unclear whether $R^i$ refers to the outcome reward or something else.
2. In PS-GRPO, although Variant 1 uses process reward, it still appears to follow the structure of an outcome reward-based method. For Variant 2, it's unclear whether the advantage term is written correctly. It’s hard to understand why the process reward in Equation (9) is multiplied by its normalized mean—this seems like it could lead to sign confusion in the rewards. Also, it seems that these two variants are not compared in the experiments?

**Ethical Concerns:**

["NO or VERY MINOR ethics concerns only"]

**Final Justification:**

The core contribution of this paper mainly lies in the promise to open-source the dataset and code. Through reading the authors' responses and discussions with other reviewers, it is found that each component of the proposed pipeline in this work is similar to existing works, with no significant innovation. Moreover, it is insufficient in terms of the magnitude of performance improvement and the expansion of theoretical depth. Therefore, I remain reserved about the possible academic influence of this paper.

**Limitations:**

Yes.

**Paper Formatting Concerns:**

1. The checklist still contains parts that should have been deleted.

2. The font size in the tables has been scaled—it's unclear whether this complies with formatting guidelines.

**Quality:**

2

**Strengths And Weaknesses:**

## Strength
1. The experiments are extensive. The ablation studies demonstrate the effectiveness of the proposed dataset MMathCoT-1M, the PRM component, and PS-GRPO.
2. Judging from the performance of the SFT model, the proposed dataset is high quality.

## Weakness
1. In the TTS evaluation, there is no comparison with the ORM trained on the same data. Therefore, it is unclear whether the improvement comes from the process reward in PRM or simply from the data used to train PRM.
2. Equation (9) and the two variants in the appendix are central to the proposed PS-GRPO method. Why are they placed in the appendix? It feels like a hack to circumvent the page limit.
3. Some typos, such as "Consistency os problematic" at line 100.

---

> ### Author Rebuttal · Authors · 2025-07-31
>
> Dear Reviewer GTaC:
>
> We sincerely thank you for your time and valuable feedback. Our responses are as follows:
>
> > Q1: Comparison with trained ORM
>
> We appreciate your constructive suggestions. We trained an ORM on the InternVL2.5-8B and URSA-8B base models using DualMath data, and the corresponding comparison has been added to the table below. Note: The URSA-8B model is used for inference.
>
> |  |   MathVerse |  |    |  | MathVista  |   |          |  | MathVision |    |     |     |
> | ------------------------ | -------- | -------- | -------- | ------------- | -------- | -------- | -------- | -------------- | -------- | -------- | -------- | -------- |
> |                          | N=4      | N=8      | N=16     | N=32          | N=4      | N=8      | N=16     | N=32           | N=4      | N=8      | N=16     | N=32     |
> | InternVL2.5-8B           | 48.6     | 50.9     | 51.8     | 51.3          | 82.5     | 83.3     | 84.3     | 85.1           | 29.9     | 32.1     | 32.8     | 33.5     |
> | InternVL2.5-ORM (trained) | 51.0     | 52.3     | 52.7     | 53.0          | 82.9     | 83.3     | 84.9     | 85.3           | 30.5     | 32.0     | 32.6     | 32.5     |
> | URSA-8B-ORM              | 52.2     | 53.4     | 53.8     | 54.3          | **83.2** | 85.0     | 85.5     | 86.5           | 30.8     | 32.1     | 33.2     | 33.9     |
> | URSA-8B-PRM              | **53.3** | **54.2** | **54.7** | **55.0**      | **83.2** | **85.5** | **86.5** | **87.2**       | **31.6** | **33.1** | **34.0** | **35.1** |
>
> * InternVL2.5-ORM (trained) is better than InternVL2.5-8B, which indicates that the data itself provides benefits.
> * The comparison on the better reasoning base model URSA-8B shows that PRM consistently outperforms ORM, which may be due to the former providing denser annotation signals.
>
> > Q2: Place formulations of two variants in Appendix
>
> Thank you for your question. As described in Lines 151-155, our two simple variants that use process rewards are intended to **reveal the issues** with integrated process rewards, which are shown in Figure 4. They function more as **pilot experiments**, demonstrating to the reader the two issues that arise from directly integrating scalar process rewards. This, in turn, motivates the description of our solution PS-GRPO (Lines 166-185). For this reason, we only described them textually in the main paper (Line 153-156) and placed their formulations in the appendix.
>
> We will make appropriate room by moving parts of the data curation section, and incorporate the formulation of the pilot experiment into the main text in revised version.
>
> > Q3: Typos
>
> Thank you for pointing out our typos. We will carefully check and correct them in the revised version. Additionally, in line 100, it should be "Consistency is problematic".
>
> > Q4: Notation Issues
>
> Thank you very much for pointing this out! We will revise the notation in Lines 664-669 according to the notation used in the main text, adopting $p$ as the annotation for process reward.
>
> $R_i$ in the equation represents the final outcome reward of a single rollout.
>
> > Q5: Clarification about two variants
>
> We will address your questions point by point:
>
> (i) Variant 1 represents an attempt to use overall process performance as the reward. Besides, given the excellent performance of rule-based outcome reward demonstrated in previous work, we considered it a harmless addition. Therefore, we incorporated it into both Variant 1 and Variant 2 to **strengthen** the baseline. This approach more effectively emphasizes the **issue** presented in Figure 4 and highlights the significance of mitigating it.
>
> (ii) We thank you for pointing out the typo in Eq. 9. The relative term is normalized to [0, 1] before multiplication. We will fix this in the revised version. The motivation for this step, inspired by vanilla GRPO, is to focus more on rollouts that are rated favorably overall by the PRM.
>
> (iii) As observed from the patterns in Figures 4 and 5, the two variants showed weaker in-domain test accuracy than PS-GRPO. For this reason, we only compared PS-GRPO against vanilla GRPO in Figure 6. However, we agree with your suggestion to include more methods that include more baselines for a direct benchmark comparison [1][2]. In addition to the two variants, we add
>
> (1) Direct process rewarding:
>
> Given a batch of process rewards: $\mathbf{R} = \\{ \\{ r_{p,1}^1, \dots, r_{p,N\_1}^1 \\}, \cdots, \\{ r_{p,1}^2, \dots, r_{p,N\_2}^2 \\} \\}$, the rewards are first normalized:
>
> $\tilde{r}\_{p,j}^i=\frac{r\_{p, j}^i - \text{mean}\\{\mathbf{R}\\}}{\text{std}\\{\mathbf{R}\\}}$,
>
> advantage of each token:
>
> $A\_{t}^i=\tilde{r}\_{p,j^*}^i,\ \  j^{\*}=\min\\{j|index(j)\geq t\\}$.
>
> Here $index(j)$ is the end token index
> of the $j$-th step.
>
> (2) DeepSeekMath's step-level GRPO (DS-GRPO).
>
> The process supervision calculates the advantage of each token as the sum of the normalized
> rewards from the following steps, i.e.:
>
> $A\_{t}^i=\sum_{index(j)>=t}\tilde{r}\_{p,j}^i$,
>
> The results comparison are show below:
>
>
> |                          | Avg      | MathVerse | MathVision | MathVista | WE-MATH  | DYNAMATH | GeoQA    |
> | ------------------------ | -------- | --------- | ---------- | --------- | -------- | -------- | -------- |
> | Variant1                 | 55.3     | 48.2      | 27.6       | 79.5      | 57.3     | 45.8     | 72.7     |
> | Variant2                 | 56.0     | 49.1      | 28.5       | 80.0      | 57.6     | 46.2     | 74.3     |
> | Direct Process Rewarding | 54.9     | 47.7      | 28.0       | 78.4      | 57.2     | 45.3     | 71.2     |
> | DS-GRPO                | 56.5     | 49.9      | 29.2       | 79.8      | 58.5     | **47.8** | 74.3     |
> | PS-GRPO (Ours)           | **58.2** | **50.9**  | **31.5**   | **83.2**  | **60.7** | 47.4     | **75.6** |
>
> These results support our claim in Lines 174–177: transforming scalar process signals into a process-to-outcome penalty provides greater robustness to length bias and reward hacking than directly using process rewards, resulting in more stable performance improvements. We ensure that the results will be added in revised version.
>
> We thank you again for your time and valuable comments. The additional experimental results you suggested help us to enhance the robustness and completeness of our claim. We believe that the added results will better demonstrate the effectiveness of URSA-8B-RM and PS-GRPO, including a more complete TTS comparison, a comparison with more online RL methods at the methodology level, and the benchmark performance of the final contributed URSA-8B-PS-GRPO.
>
> If you have any other questions, feel free to discuss further!
>
> [1] Wang, Peiyi, et al. Math-Shepherd: Verify and Reinforce LLMs Step-by-step without Human Annotations
>
> [2] Shao, Zhihong, et al. Deepseekmath: Pushing the limits of mathematical reasoning in open language models

---

> ### Author Response · Authors · 2025-08-08
>
> Dear Reviewer GTaC:
>
> We thank you for your prompt response, and we appreciate your recognition of our work. Regarding the contribution of the proposed pipeline, we will provide the following explanation.
>
> > Engineering and Algorithm Contribution
>
> **Data and model contribution**: As motivated in our Introduction (lines 33-37), we are dedicated to solving the questions of "on which models is process supervision more valuable" and "how to create multimodal process supervision data." Ultimately, we use MMathCoT to demonstrate that CoT scaling can improve the base model's pass@N (Figure 7) while also raising the upper limit for the application of process supervision. During the creation of the DualMath-1.1M dataset, although we drew on the common practice of MCTS value-guided data synthesis from unimodal reasoning, we emphasize that in the unique multimodal context, focusing on data expansion for image-text inconsistency is an effective, non-sampling-dependent means of **scaling process supervision data** (Table 3, 10).
>
> Our generalization experiments show that URSA-8B-RM demonstrates robustness for TTS and RL on models of equivalent size, such as InternVL2.5-8B (Table 2, 8, 10). This indicates that we have established a **relatively strong data and model foundation for research into multimodal PRMs**. To our knowledge, we are the **first** to accomplish this. In the absence of exploration in non-concurrent work, this foundation is a necessary accomplishment.
>
> **PS-GRPO**: In fact, the significance of PS-GRPO lies in its direct reversal of the performance trends of methods using scalar process rewards for online RL (Figures 4, 5). The results are encouraging for exploring the paradigm of **combining process rewards and outcome rewards**. We have proposed two key insights for the exploration of PRM in RL:
>
> (i) In addition to causing reward hacking, the use of a scalar process reward can lead to phased reward bias, ultimately resulting in shortened inference length (Figure 4).
>
> (ii) The failure mode of PRM-integrated online RL does not lie in the PRM's training method, as its supervisory capability does not diminish with the base model's online training (Figure 5).
>
> Newly added baselines, such as DS-GRPO, do not escape the first claim. Based on these two key findings, we use a formally simple but effective transformation to convert the sequential process reward, which is prone to being hacked and has length bias, into a robust reward penalty. This directly achieves the goal of **mitigating the issue and improving benchmark performance**. This process-to-outcome penalty design does not overlap with existing work.
>
> > Performance gain and theory
>
> In fact, When compared to non-concurrent work, URSA-8B and URSA-8B-PS-GRPO show an average advantage of **4.9** and **8.4** points, respectively, over the strongest similarly-sized baseline, Gemma3-12B, across six benchmarks. Compared to concurrent work as defined by NeurIPS 2025, strong baselines using the same SFT+RL paradigm, such as Vision-R1 [1] and MM-Eureka [2], still show a performance disadvantage of 1.8 and 2.4 points on average compared to URSA-8B-PS-GRPO.
>
> In terms of RL method comparison, PS-GRPO shows an average performance advantage of 3.3 and 1.7 points over direct process reward and DS-GRPO, respectively. Meanwhile, recent works focusing on advantage design, GSPO [3] and GMPO [4], only manage to surpass vanilla GRPO by **1.2** and **1.1** points on our six benchmarks. This further demonstrates that the paradigm we propose for exploring the combination of outcome and process rewards offers a valuable path, and we have provided a foundation for mitigating the apparent issues.
>
> For the theoretical explanation, as in our response to reviewer U6mC, we hope to demonstrate from the Q-value ranking theorem that PS-GRPO is an effective proxy for ideal process supervision score modeling.
>
> In summary, please allow us to appropriately describe our workload. URSA not only involves the data and model research foundation from the RFT to RL, but we also propose a new reward modeling path for the RL stage. In addition to the final benchmark comparisons, we select appropriate baselines for TTS and RL methods, as well as generalization validations.
>
> We fully respect your evaluation decision, it is your suggestions that have made our paper more complete. We are also grateful that you are willing to take the time to participate in this discussion, which makes us feel respected. We promise that the content based on your suggestions will be added to revised version.
>
> [1] Vision-r1: Incentivizing reasoning capability in multimodal large language models. Arxiv 2503.
>
> [2] MM-Eureka: Exploring Visual Aha Moment with Rule-based Large-scale Reinforcement Learning. Arxiv 2503.
>
> [3] Qwen Team. Group Sequence Policy Optimization. Arxiv 2507.
>
> [4] Geometric-Mean Policy Optimization. Arxiv 2507.

---

### Official Review · Reviewer_4akc · 2025-07-01

**Clarity:** 4
**Significance:** 2
**Originality:** 2
**Rating:** 4
**Confidence:** 4

**Summary:**

This paper presents URSA, a three-stage framework to enhance multimodal mathematical reasoning with Multimodal Large Language Models (MLLMs) by leveraging Process Reward Models (PRMs). The authors address key challenges including the lack of high-quality multimodal reasoning data, insufficient automated process labeling methods, and issues in reward modeling during reinforcement learning (RL). They propose (1) MMathCoT-1M, a large-scale multimodal CoT dataset; (2) DualMath-1.1M, a process-supervised dataset to train PRMs; and (3) PS-GRPO, a reinforcement learning method that integrates PRMs via relative process supervision. Experiments on six multimodal benchmarks show competitive results, outperforming strong open-source and commercial models.

**Questions:**

1. CoT Dataset Quality: From the Appendix examples, the constructed CoT chains in MMathCoT-1M appear relatively short and lack higher-order reasoning features like reflection or verification. Can the authors provide a more systematic quantitative analysis (e.g., distribution of step counts, presence of verification tokens) and clarify why these CoTs are suitable for warming up reasoning?

2. Novelty over Existing PRM Methods: The binary search-based PRM training pipeline is very similar to OmegaPRM. Could the authors elaborate on what concrete differences exist (if any), especially in reward signal generation or sampling strategies?

3. Comparison with Step-Level GRPO: PS-GRPO closely resembles the step-level GRPO strategy proposed in DeepSeekMath. Why is there no detailed empirical or conceptual comparison? Including this comparison seems necessary to clarify the technical novelty of PS-GRPO.

4. Baseline Selection: Since the paper claims state-of-the-art performance among open-source MLLMs, it would strengthen the claim to include more competitive baselines like Multimodal Open R1-8k-verified or other recent strong MLLMs that perform math reasoning with CoT.

**Ethical Concerns:**

["NO or VERY MINOR ethics concerns only"]

**Final Justification:**

The core contribution of this paper lies primarily in engineering—transferring language-model reasoning paradigms into the visual modality. After reading the rebuttal, I still do not see standout algorithmic or theoretical innovations, as most components resemble existing approaches in data construction, PRM training, and RL fine-tuning. Nevertheless, the experiments demonstrate reasonable effectiveness and some integration value, which led me to slightly raise my score.

That said, the improvements over concurrent works are not striking, and I remain somewhat cautious about the paper’s overall impact.

**Limitations:**

yes

**Quality:**

2

**Strengths And Weaknesses:**

Strengths:

1. The paper targets a timely and underexplored direction: integrating PRMs into multimodal mathematical reasoning.

2. Experimental results across six diverse benchmarks demonstrate strong performance, surpassing models like GPT-4o and Gemma3-12B.

3. The proposed PS-GRPO method shows empirical gains and addresses known issues in scalar PRM reward modeling, such as reward hacking and length bias.

Weaknesses:

1. The constructed CoT dataset (MMathCoT-1M) lacks depth and complexity; examples in the appendix indicate short chains without reflective or verification steps, which may limit its effectiveness as a reasoning warm-up corpus.

2. The PRM training process closely follows OmegaPRM’s binary search approach, with limited technical innovation.

3. The proposed PS-GRPO algorithm is similar in spirit to DeepSeekMath’s step-level GRPO, but no discussion or comparison is provided.

4. The training paradigm (warm-up then RL) is standard for reasoning models, yet key baselines like Multimodal Open R1 are missing, which weakens the significance of reported gains.

The originality across each stage of the pipeline is limited; the contribution lies more in system integration than in novel algorithmic designs.

---

> ### Author Rebuttal · Authors · 2025-07-31
>
> Dear Reviewer 4akc:
>
> We sincerely appreciate your insightful comments. We respond to your questions point by point below.
>
> > Q1: The constructed CoT dataset (MMathCoT-1M) lacks depth and complexity
>
> Thank you for your question. We first provide a quantitative analysis and make direct comparisons with Mulberry-SFT [1], Vision-R1-ColdStart [2], and R1-OneVision-ColdStart [3]. Then, we illustrate the advantages of our data from three perspectives.
> * Direct Comparison: For data without explicit "Step X" labels, we use '\n\n' as the step separator. For the reflection pattern, we follow the lexical statistics convention from the Vision-R1[2], and the results are shown in the table below:
>
>     |                      | Mulberry-SFT | Vision-R1-ColdStart | R1-OneVision-ColdStart | MMathCoT (Ours) |
>     | -------------------- | ------------ | ------------------- | ---------------------- | --------------- |
>     | Average Step Number  | 5.3          | 7.3                 | 7.5                    | 6.4             |
>     | Average Token Length | 439          | 887                 | 921                    | 488             |
>     | Word "Mistake"       | 8,784        | 26,697              | 14,432                 | 11,327          |
>     | Word "Alternatively" | 28           | 188,187             | 101,043                | 29,549          |
>     | Word "Check"         | 26,421       | 100,148             | 55,873                 | 75,653          |
>
> From the perspective of reasoning steps, MMathCoT does not show significant differences compared to the two datasets synthesized by combining Caption + DeepSeek-R1. The main differences lie in token length and vocabulary related to reflection. However, some methods focusing on fast-slow thinking suggest that response length is not entirely correlated with effectiveness [4], and we will demonstrate this point with empirical results later.
> Reason for using our data:
> * Empirical results: We noticed that in the Vision-R1 and R1-OneVision papers, using coldstart data for SFT resulted in average performance that was worse or on par compared to their Qwen2.5-VL base model. We used these two datasets instead of MMathCoT for continual pretraining on the aligned URSA-8B, and the results are as follows (we set max new token to 32768 in vLLM):
>
>     | Dataset      | Avg Acc |
>     | ------------ | ------- |
>     | Vision-R1    | 51.9    |
>     | R1-OneVision | 48.7    |
>     | MMathCoT     | 54.7    |
>
>     We suspect that a possible reason is that the data generated by caption + R1-like LLM focuses more on reflecting on textual caption or generated tokens rather than truly achieving reflection on image feature. Therefore, we achieved similar conclusions to those in paper [4] under similar token costs. Besides, we provide a clear scaling law in Table 9, which demonstrates the effectiveness of the continual pretraining stage.
> * Length-triggered regularity: In Figure 5(c), our URSA-8B trained with MMathCoT also shows length-induced pattern when applying vanilla GRPO and PS-GRPO. And finally, we achieve better average performance with less token cost.
>
>     |                 | Avg. Token Length | Avg. Acc |
>     | --------------- | ----------------- | -------- |
>     | URSA-8B-PS-GRPO | 471               | **58.2** |
>     | Vision-R1       | 899               | 56.4     |
>     | R1-OneVision    | 934               | 52.2     |
>
> > Q2: Compare with OmegaPRM
>
> We acknowledge the similarity in using binary search-based localization, as adopted by OmegaPRM. However, our contribution lies in extending this to the multimodal setting, where visual-textual inconsistency presents unique challenges. To this end, we propose the Misinterpretation Insertion Engine (MIE) for hallucination-oriented supervision, combined with Binary Error Locating (BEL), to construct the DualMath-1.1M dataset. The effectiveness of both components is validated through ablations in Table 3.
>
> > Q3: Compare with DeepSeekMath's step-level GRPO
>
> Thanks for pointing out that DeepSeek Process Supervised GRPO (hereafter DS-GRPO) should be included in the comparison. We respectfully clarify that DS-GRPO is best viewed as a complement to our pilot experiment in Figure 4, as their core motivations differ. Below, we analyze the comparison from two perspectives:
> * We claim that DS-GRPO, just like the two variants we used in the pilot experiment, still directly uses the scalar reward generated by PRM to calculate advantages. On the other hand, our PS-GRPO is the first to use process-to-outcome reward penalty. Specifically:
>     * Variant 1: Uses the average process reward of the rollout as the final reward. (Line 666)
>     * Variant 2: Reweights the reward of each step by its relative advantage over the in-batch mean process reward. (Equ. 9)
>     * DS-GRPO: Calculates step-level advantage by considering all in-batch process rewards. Each token's advantage is the sum of rewards from all subsequent steps.
>
>     All of these are variants that leverage **scalar process rewards** to assist online RL. From the perspective of Figure 4, we conducted experiments on DS-GRPO and were pleased to find that it does outperform the two variants we tested in terms of response length and in-domain test accuracy. However, it still suffers from **length decay** and **reward hacking**. Therefore, the inclusion of DS-GRPO in the discussion does not alter the claim made in Lines 176-177.
> * We provide a direct benchmark comparison of Variant 1, Variant 2, DS-GRPO, and PS-GRPO (Ours) on the URSA-8B model in the table below.
>
>     |                | Avg      | MathVerse | MathVision | MathVista | WE-MATH  | DYNAMATH | GeoQA    |
>     | -------------- | -------- | --------- | ---------- | --------- | -------- | -------- | -------- |
>     | Variant1       | 55.3     | 48.2      | 27.6       | 79.5      | 57.3     | 45.8     | 72.7     |
>     | Variant2       | 56.0     | 49.1      | 28.5       | 80.0      | 57.6     | 46.2     | 74.3     |
>     | DS-GRPO        | 56.5     | 49.9      | 29.2       | 79.8      | 58.5     | **47.8** | 74.3     |
>     | PS-GRPO (Ours) | **58.2** | **50.9**  | **31.5**   | **83.2**  | **60.7** | 47.4     | **75.6** |
>
> Therefore, we demonstrate the uniqueness of PS-GRPO in alleviating reward length bias and reward hacking, and provide empirical advantages over direct scalar process reward utilization.
>
> > Q4: R1-like baseline
>
> Thank you for pointing this out. Since the vast majority of public R1-style reasoning MLLMs were released after March 1st, 2025, they are considered concurrent work under NeurIPS 2025 policy and were thus not included in our main table.
>
> Your suggestions help further demonstrate the value introduced by the proposed PRM-integrated RL. We have placed the results comparison in the table below. The official implementation of Multimodal-Open R1-8k-verified is based on Qwen2-VL, and we additionally include results on the more capable Qwen2.5-VL model for completeness.
>
> |                               | Avg      | MathVision | MathVerse | MathVista | WE-MATH  | DYNAMATH | GeoQA    |
> | ----------------------------- | -------- | ---------- | --------- | --------- | -------- | -------- | -------- |
> | Qwen2.5-VL+OpenR1-8K-verified | 53.1     | 25.7       | 48.9      | 72.9      | 56.3     | 47.4     | 65.9     |
> | OpenVLThinker-7B              | 53.8     | 25.9       | 50.3      | 76.4      | 59.2     | 46.6     | 64.5     |
> | Adora-7B                      | 54.6     | 25.5       | 50.1      | 77.6      | 58.4     | 45.9     | 70.1     |
> | R1-OneVision                  | 52.2     | 29.9       | 46.4      | 69.7      | 57.2     | 46.5     | 64.6     |
> | MM-Eureka-7B                  | 55.8     | 26.9       | 50.3      | 76.5      | 60.2     | **49.1** | 70.9     |
> | Vision-R1                     | 56.4     | 28.0       | **52.4**  | 82.4      | 58.8     | 48.0     | 68.6     |
> | URSA-8B-PS-GRPO               | **58.2** | **31.5**   | 50.9      | **83.2**  | **60.7** | 47.4     | **75.6** |
>
> We find that compared to concurrent r1-like MLLMs, URSA-8B-PS-GRPO still achieves the best average score.
>
> > Q5: System contribution
>
> While we appreciate your perspective, we wish to clarify that our data contribution is indispensable, particularly for non-concurrent work where no open-source multimodal PRM is available.
>
> Our contributions are multifaceted across the stages:
> * In Stages 1 & 2, we pioneered novel data synthesis methodologies, performed essential ablation analyses to demonstrate their impact, and released large-scale datasets to advance the research foundation.
> * In Stage 3, our approach is distinct from prior work that simply applies methods like GRPO, REINFORCE. First, we explicitly dissect the failure modes of various approaches that use direct scalar process rewards in multimodal RL. Second, we pioneer a process-to-outcome framework designed to mitigate two prominent issues in pilot experiments. The innovation and effectiveness of our PS-GRPO method are empirically validated by the results in Figure 6 and the additional experiments provided.
>
> > Q6-Q9
>
> Please see our responses to Q1-Q4, respectively.
>
> We sincerely appreciate your time and feedback. We confirm that the additional comparisons and discussions will be included in the revised version. If you have any further questions, please feel free to let us know.
>
> [1] Yao, Huanjin, et al. Mulberry: Empowering MLLM with o1-like Reasoning and Reflection via Collective Monte Carlo Tree Search
>
> [2] Huang, Wenxuan, et al. Vision-R1: Incentivizing Reasoning Capability in Multimodal Large Language Models
>
> [3] Yang, Yi, et al. R1-Onevision: Advancing Generalized Multimodal Reasoning through Cross-Modal Formalization
>
> [4] Xiao, Wenyi, et al. Fast-Slow Thinking for Large Vision-Language Model Reasoning

---

> > ### Comment · Reviewer_4akc · 2025-08-07
> >
> > The core contribution of this paper lies primarily in engineering—transferring language-model reasoning paradigms into the visual modality. After reading the rebuttal, I still do not see standout algorithmic or theoretical innovations, as most components resemble existing approaches in data construction, PRM training, and RL fine-tuning. Nevertheless, the experiments demonstrate reasonable effectiveness and some integration value, which led me to slightly raise my score.

---

> ### Author Response · Authors · 2025-08-07
>
> Dear Reviewer 4akc:
>
> We are very grateful for your prompt reply and appreciate your willingness to raise the overall evaluation of our work. Next, please allow me to briefly summarize the contributions made to the work.
>
> First, we fully agree with your point that our work indeed covers a significant amount of engineering implementation. Our focus in Stages 1 & 2 on how to train a good multimodal PRM and provide a solid foundation of data and model research to community is a widely recognized necessity [1][2][3].
>
> Building upon this foundation, we were not content to validate its value only by BoN, as is common in other PRM-related work [4][5]. Instead, we **uncovered and challenged the dilemma of integrating process rewards and outcome rewards in online RL**.
>
> We have provided some insightful findings:
>
> (i) in addition to causing reward hacking, the use of a scalar process reward can lead to phased reward bias, ultimately resulting in shortened inference length (Figure 4);
>
> (ii) the failure mode of PRM-integrated online RL does not lie in the PRM's training method, as its supervisory capability does not diminish with the base model's online training (Figure 5).
>
> These motivate us using a **formally simple but effective** method, which not only surpasses the vanilla rule-based outcome GRPO but also mitigates the issues with process supervision.
>
> We are very thankful for the valuable suggestions you provided, which has greatly helped us improve our work. We are also grateful for your kind reply, which has given us a great deal of positive feedback. We will ensure that the supplementary content will be fully incorporated into the revised version.
>
> [1] G-llava: Solving geometric problem with multi-modal large language model. ICLR25
>
> [2] Mavis: Mathematical visual instruction tuning with an automatic data engine. ICLR25
>
> [3] Math-PUMA: Progressive Upward Multimodal Alignment to Enhance Mathematical Reasoning. AAAI25
>
> [4] Qwen Team. The Lessons of Developing Process Reward Models in Mathematical Reasoning. Arxiv 25.01
>
> [5] rStar-Math: Small LLMs Can Master Math Reasoning with Self-Evolved Deep Thinking. ICML25

---

### Official Review · Reviewer_U6mC · 2025-07-03

**Clarity:** 3
**Significance:** 2
**Originality:** 2
**Rating:** 3
**Confidence:** 4

**Summary:**

This paper introduces URSA, a three-stage framework for enhancing multimodal mathematical reasoning through Process Reward Models (PRMs). The authors address the challenge of applying PRMs, which have shown success in text-only LLMs, to multimodal contexts. The framework consists of: (1) creating MMathCoT-1M, a large-scale multimodal Chain-of-Thought dataset to train a stronger foundation model URSA-8B, (2) constructing DualMath-1.1M via a dual-view process supervision strategy that addresses both logical validity and visual grounding, and (3) proposing PS-GRPO (Process-Supervised Group-Relative-Policy-Optimization), a novel RL method that mitigates reward hacking and length bias issues inherent in direct PRM application. The resulting URSA-8B-PS-GRPO model achieves state-of-the-art performance among open-source MLLMs, outperforming Gemma3-12B by 8.4% and GPT-4o by 2.7% on average across 6 benchmarks.

**Questions:**

1. Can you provide any theoretical justification for why the "drop-moment" threshold ρ=0.3 works well? Have you explored adaptive thresholds based on problem difficulty?
2. What percentage of your automatically generated process labels were manually verified? How confident are you that the synthetic misinterpretation insertion captures real failure modes in multimodal reasoning?

**Ethical Concerns:**

["NO or VERY MINOR ethics concerns only"]

**Limitations:**

Yes, the authors have adequately addressed limitations in Section I and throughout the paper. They acknowledge the computational requirements, focus on STEM tasks, and potential biases in synthetic data generation. However, they could expand on:

- The potential for PRMs to reinforce systematic biases in the training data
- Failure modes when visual and textual information strongly conflict
- Scalability challenges for even larger multimodal models

**Quality:**

2

**Strengths And Weaknesses:**

Strengths
1. Interesting Framework: The three-stage approach is well-motivated, addressing data scarcity, automated process labeling, and reward modeling issues in a pretty principled manner. PS-GRPO is a clever solution to reward hacking, using "drop-moments" in PRM sequences to penalize rollouts with correct outcomes but flawed reasoning processes.
2. Good Empirical Results: The method achieves impressive performance gains across multiple benchmarks, with particularly notable improvements on challenging datasets like MathVision (31.5 vs 30.4 for GPT-4o).
3. Public benchmarks: The release of two large-scale datasets (MMathCoT-1M and DualMath-1.1M) will be valuable for the community.

Weaknesses
1. Cost: The paper requires significant resources (e.g., 2.7M API calls to Gemini-1.5-Flash).
2. Limited Theoretical Justification: While PS-GRPO works empirically, the paper lacks theoretical analysis of why penalizing "drop-moments" is optimal or under what conditions this approach might fail.
3. Process Label Quality: The automated process labeling relies heavily on Monte Carlo estimation and synthetic error insertion, which may introduce biases not fully explored.

---

> ### Author Rebuttal · Authors · 2025-07-31
>
> Dear Reviewer U6mC:
>
> Thank you for your thorough review and constructive feedback.
>
> > Q1: High cost issue
>
> We understand your concerns and will address this issue from two aspects. (i) We compare our work with other multimodal reasoning dataset below (per million tokens).
>
> ||Size|Model|Input token price|Output token price|
> | --------------- | ---- | ---------------- | -------------------------------------- | --------------------------------------- |
> | Ours            | 2.1M | Gemini-1.5-Flash | 0.07$                                  | 0.3$                                    |
> | LLaVA-CoT       | 100K | GPT-4o           | 2.7$                                   | 10$                                     |
> | Mulberry        | 260K | GPT-4o           | 2.7$                                   | 10$                                     |
> | VisualRationale | 193K | GPT-4o           | 2.7$                                   | 10$                                     |
>
> Our total cost is actually lower. We conduct a comprehensive comparison with GPT-4o in Appendix E.2, which demonstrates that our choice is **cost-effective**. And our total cost is statistically around $320, which is a relatively modest amount.
> (ii) We are fully committed to open community. All data are publicly released via the provided link. We believe providing high-quality, open resources justifies the cost and is essential for advancing the research community.
> > Q2: Theoretical analysis
>
> Thank you for your question. Our theoretical explanation is grounded in the Q-value ranking theorem [1].
>
> **Theoretical Background**: In a multi-step reasoning Markov Decision Process, the optimal Q-function, $Q^*(s_t, a_t)$, is the expected probability of reaching the correct final answer after action $a_t$. The scalar reward $r_p$ from our URSA-8B-RM at each step serves as an empirical estimate, $Q_{\text{est}}$, of this true Q-value.
>
> **Theoretical Optimal Objective**: Approximating the Ideal Q-Value Ordering
> According to Theorem 3.5 in PQM, under an optimal reasoning policy $\pi^* $, the Q-values along a reasoning path should satisfy a strict monotonic ordering:
> $Q^*_{w\_{|W|}}<\cdots<Q^{\*}\_{w_1}\ll V^{\*}(x)<\cdots<Q^{\*}\_{c_1}<Q^{\*}\_{c\_{|C|}}$, where $c_i$ and $w_i$ denote the i-th correct step and the i-th incorrect step, respectively.
> In our drop-moment definition, $c\_{|C|}$ is adjacent to $w\_1$, which is consistent. The ideal training objective for a PRM is to learn a Q-estimator $Q_{\text{est}}$ that strictly follows this ordering across all reasoning paths. For a successful reasoning path $\tau$(i.e., $o(\tau)=1$), this implies that the sequence of $Q\_{\text{est}}$ outputs $\{r_{p,1}, r_{p,2}, \ldots, r_{p,T}\}$ should be monotonically increasing with no decreases, as a successful path should not contain any error.
>
> Thus, the theoretically optimal generation objective is:
> $J\_{\text{ideal}}(\theta) = \mathbb{E}\_{\tau \sim \pi\_\theta} [ o(\tau) ] - \lambda \cdot \Omega(G(\tau))$
>
> where:
>
> * $\mathbb{E}\_{\tau \sim \pi\_\theta} [ o(\tau) ]$ is the standard final result accuracy.
>
> * $\Omega(G(\tau))$ is a non-decreasing regularization term measuring the violation of the ideal Q-value ordering. $\lambda$ is its strength.
>
> $G(\tau) = r_{c_{|C|}}(\tau) - r_{w_1}(\tau)$ is a direct measure of the violation of the theoretical gap. The larger $G(\tau)$, the greater the drop from high (correct) to low (incorrect) rewards at some point in the path, which severely violates the ideal ordering defined in Theorem 3.5. Thus, minimizing $G(\tau)$ is equivalent to aligning the model's generation process with the Q-value dynamics of the optimal reasoning path as defined by PQM.
>
> **PS-GRPO as a Proxy Implementation**: PS-GRPO introduces a computable, differentiable proxy for $\Omega(G(\tau))$ into its reward function:
> $R(\tau) = o(\tau) \cdot (1 - \gamma \cdot \mathbf{I}(\delta_p(\tau) \geq \rho))$
>
> This function approximates the ideal objective $J_{\text{ideal}}(\theta)$:
> * The $o(\tau)$ term directly corresponds to the first term in $J_{\text{ideal}}$, maximizing final accuracy.
> *   The penalty term approximates the regularizer:
>     * $\delta_p(\tau)$ is a computationally tractable and strongly correlated proxy for $G(\tau)$. A large $\delta_p$ indicates a large $G$.
>     * The indicator function $\mathbf{I}(\delta_p(\tau) \geq \rho)$ converts the continuous $\delta_p$ into a binary signal, resolving non-differentiability and ensuring compatibility with GRPO.
>     * The hyperparameter $\rho$ acts as the violation threshold for $\Omega(G(\tau))$ and is determined by the behavior of $Q_{\text{est}}$.
>
> Therefore, the PS-GRPO reward function $R(\tau)$ serves as a practical proxy for $J_{ideal}$: $J_{\text{PS-GRPO}}(\theta) \approx J_{\text{ideal}}(\theta)$
>
> The final step is to empirically determine $\rho$ for a given PRM. As mentioned in Q4, we detail this process in a follow-up response.
>
> > Q3: Data synthesis biases
>
> Thank you for your question. We address potential biases in our two subsets.
>
> **BEL**: Bias primarily depends on the inference model, stronger base models always yield higher-quality MCTS annotations. We validate BEL’s reliability via:
>
> 1. Generalization: The w/o MIE checkpoint (Table 3) generally outperforms baselines, indicating robust generalization
> |                |                    | MathVerse |      | MathVista |      | MathVision |      |
> | -------------- | ------------------ | --------- | ---- | --------- | ---- | ---------- | ---- |
> |                |                    | N=8       | N=32 | N=8       | N=32 | N=8        | N=32 |
> | InternVL2.5-8B | SC                 | 42.2      | 42.6 | 66.9      | 67.6 | 21.7       | 22.0 |
> |                | InternVL2.5-8B ORM | 42.1      | 42.7 | 67.5      | 67.9 | 21.9       | 22.5 |
> |                | URSA-8B-RM w/o MIE | 44.1      | 46.2 | 68.7      | 69.4 | 22.9       | 23.2 |
> | EMOVA          | SC                 | 46.7      | 47.3 | 77.9      | 79.0 | 26.8       | 28.0 |
> |                | InternVL2.5-8B ORM | 45.6      | 46.1 | 77.7      | 79.2 | 26.6       | 27.8 |
> |                | URSA-8B-RM w/o MIE | 48.2      | 48.0 | 78.3      | 79.5 | 27.4       | 29.2 |
>
> 2. LLM-as-a-Judge [2-3]: On 2,000 random samples, Gemini 2.5 Pro shows 87.6% consistency in first-error identification, confirming annotation quality.
>
> **MIE**: As shown in Appendix G.2, MIE's construction involved extensive prompt engineering and manual review. We demonstrate its impact from two perspectives:
>
> 1. LLM-as-a-Judge: Gemini 2.5 Pro achieves 95.9% consistency on 2,000 random samples. We attribute this agreement to Gemini 2.5 Pro's strong perceptual capabilities and sensitivity to visual inconsistencies.
>
> 2. Ablation: Replacing MIE with BEL data of same scale diminishes performance, confirming MIE’s gain stem from new critical signals, not data volume—indicating its value in TTS is robust to potential bias.
>
> |               | MathVerse |          | MathVista |          | MathVision |          |
> | ------------- | ------------- | -------- | ------------- | -------- | -------------- | -------- |
> |               | N=8           | N=32     | N=8           | N=32     | N=8            | N=32     |
> | PRM800K       | 50.8          | 51.8     | 82.7          | 83.2     | 30.6           | 31.4     |
> | Expanded BEL  | 53.5          | 54.2     | 84.7          | 85.9     | 32.7           | 34.4     |
> | DualMath-1.1M | **54.2**      | **55.0** | **85.5**      | **87.2** | **33.1**       | **35.1** |
>
> > Q4: Value of ρ and adaptive threshold
>
> The optimal threshold for a specific PRM varies with the dataset and task. To define a reasonable range for our URSA-8B-RM, we follow Figure 4 in [1], testing on 30K false-positive rollouts (correct outcomes with flawed reasoning). Our analysis identified an $\delta_p^i$ range of around 0.25–0.6, validating the parameter search in Table 4. For comparison, solutions with entirely correct process reasoning yield a much lower $\delta_p^i$ range, averaging below 0.08.
>
> Following your insightful suggestion, we explore a dynamic approach. This method adjusts $\rho$ based on in-batch difficulty:
> * For difficult problems (rollout accuracy ≤ 0.25), we use a more lenient $\rho$ = 0.5.
> * For easy problems (rollout accuracy ≥ 0.75), we apply a stricter $\rho$ = 0.2.
> * For all other cases, ρ is set to 0.3.
> Encouragingly, the dynamic approach shows positive results, as shown below.
>
> |                        | Avg  | MathVerse | MathVision | MathVista | WE-MATH | DYNAMATH | GeoQA |
> | ---------------------- | ---- | --------- | ---------- | --------- | ------- | -------- | ----- |
> | w/o adaptive ρ | 58.2 | 50.9      | 31.5       | 83.2      | 60.7    | 47.4     | 75.6  |
> | w/ adaptive ρ  | 58.8 | 51.6      | 32.2       | 82.6      | 61.7    | 48.5     | 76.4  |
>
> This indicates that our novel process-to-outcome reward modeling has extensibility and offers room for further exploration. We will include it in the revised version.
>
> > Q5: Manual verification and synthetic data quality
>
> We manually verified 300 samples during prompt engineering in Appendix G.2, requiring each corrupted instance to have a plausible error (acceptance threshold for prompt is 90%). Our confidence is grounded in two aspects: (1) its error typology is derived from prior work, ensuring theoretical breadth [4-5]; and (2) Table 3 and the response to Q3 show MIE-trained PRMs effectively identify image-text inconsistency, illustrating empirical validity.
>
> Thank you once again. Please feel free to discuss further!
>
> [1]  PROCESS REWARD MODEL WITH Q-VALUE RANKINGS. ICLR25
>
> [2] The Lessons of Developing Process Reward Models in Mathematical Reasoning
>
> [3] LLM Reasoners: New Evaluation, Library, and Analysis of Step-by-Step Reasoning with Large Language Models
>
> [4] ErrorRadar: Benchmarking Mathematical Reasoning of Multimodal Large Language Models Via Error
>
> [5] VisOnlyQA: Large Vision Language Models Still Struggle with Visual Perception of Geometric Information

---

> > ### Author Response · Authors · 2025-08-06
> >
> > Dear Reviewer U6mC,
> >
> > We would like to express our sincere gratitude for your constructive comments. Following your suggestions, we have done our best to implement additional experiments.
> >
> > We are writing to kindly follow up on our rebuttal. As there are just over two days remaining in the discussion period, we are keen to ensure we have fully addressed your concerns. We would greatly appreciate hearing your thoughts on our response and the new results. This would give us the opportunity to offer more details or clarifications if anything remains unclear.
> > Thank you again for your time and guidance.
> >
> > Best regards,
> >
> > The Authors of Submission 5507

---

> ### Author Response · Authors · 2025-08-08
> **Follow-up on Discussion**
>
> Dear Reviewer U6mC,
>
> I hope this message finds you well. As the discussion period is nearing its end with less than one days remaining. I want to ensure we have addressed all your concerns satisfactorily. If there are any additional points or feedback you'd like us to consider, please feel free to let us know. Your insights are invaluable to us, and we're eager to address any remaining issues to improve our work.
>
> Thank you for your time and effort in reviewing our paper.
>
> Best regards,
>
> Authors of Submission 5507

---

### Official Review · Reviewer_E38C · 2025-07-16

**Clarity:** 3
**Significance:** 2
**Originality:** 2
**Rating:** 5
**Confidence:** 2

**Summary:**

The authors address prior shortcomings in the application of process reward models (PRMs) to multimodal mathematical reasoning models. Specifically, they introduce URSA, a 3-stage training framework that rests on the following steps:

- They build a novel math reasoning foundation MLLM (URSA-8B) by training a LLaVA-like architecture on a novel dataset MMathCoT-1M compiled by combining and filtering a number of pre-existing datasets
- URSA-8B is then turned into URSA-8B-RM by finetuning on a novel process supervision dataset, DualMath-1.1M, which is enhanced through first error step identification (BEL) and an misinterpretation insertion engine (MIE).
- The final stage fixes existing issues with integrating process reward models (PRM) into RL post-training. The authors overcome these by introducing PS-GRPO, which uses a reward thresholded by a newly-introduced "drop-moment".

The authors evaluate their method innovations on a number of of multimodal math reasoning benchmarks and achieve state-of-the-art results for open-source MLLMs, and also present ablations into BEL and MIE.

**Questions:**

- Could you provide confidence intervals (e.g. bootstrapped std-of-mean) for you main empirical results (Table 1)?
- Table 1: Why are you employing different metrics for different datasets?
- How sensitive is the "drop-moment" approach to hyperparameter selection?
- Section 6.2: Can you clarify that including BEL/MIE doesn't simply lead to performance improvement as more training samples are included/more training steps are executed?

**Ethical Concerns:**

["NO or VERY MINOR ethics concerns only"]

**Final Justification:**

The authors have addressed various empirical and theoretical concerns adequately, however, I remain unsure about the conceptual innovativeness and novelty aspect of this paper.

**Limitations:**

yes

**Quality:**

3

**Strengths And Weaknesses:**

# Strengths

- The quality of the paper is high overall, it is written clearly and the empirical results appear statistically significant*
- The methods introduced appear previously unpublished, and are mechanistically credibly leading to performance improvement
- The inclusion of ablation studies on BEL and MIE is commended, as is the analysis of PRM shortcomings (Section 4)

# Weaknesses

- While the paper results in additional practical "tricks" that appear to lead to improved reasoning performance, it does often remain unclear how sensitive the introduced methods are to hyperparameter tuning and whether their specific incarnation is justified (e.g. the concept of a "drop-moment")
- *we don't have confidence intervals for empirical results (e.g. variance/std of the mean)

---

> ### Author Rebuttal · Authors · 2025-07-31
>
> Dear Reviewer E38C:
>
> Thank you very much for your time and insightful suggestions. We provide our point-by-point responses below.
>
> > Q1: Hyperparameter sensitivity and rationale of methodology.
>
> Thank you for your question. We will address each point below.
> * We appreciate your attention to PS-GRPO’s robustness. As presented in Section 6.3 and Table 4, we performed a **sensitivity analysis** on its two core hyperparameters: the reward penalty $\gamma$ and the drop-moment threshold $\rho$.
>
>     We consider $\rho$ to be a relatively more sensitive parameter, as it also affects the PRM's effectiveness in first error identification during the online RL process (see Figure 5(b)). For a fixed $\rho$ (e.g., 0.3), the choice of $\gamma$ is relatively robust within a reasonable range. For instance, using $\gamma$ = 0.3, 0.5, 0.7 yields average performances of 57.9, 58.2, and 57.5 respectively. Therefore, applying PS-GRPO primarily requires choosing an appropriate $\rho$ for a specific PRM, while $\gamma$ is easier to select in practice.
>
>     Meanwhile, the value of $\rho$ can be determined by performing batch statistics on in-domain false positive data to obtain the search bounds.
>
> * Regarding the rationale of PS-GRPO and the "drop-moment," we would like to reinforce our explanation by referencing Figure 5 (a) and (b). As discussed in Line 174-177, our insight is that while the direct use of a scalar process reward can lead to reward hacking, the ability of a well-trained PRM to distinguish the quality of online rollouts on its base model does not degrade as training progresses. Therefore, we leverage this experimental observation to transform the scalar reward into a scenario-specific outcome reward penalty.
>
>     Furthermore, our method was not validated solely on the URSA-8B model. In Appendix C.4, we conducted generalization experiments on the InternVL2.5-8B and MultiMath models, where PS-GRPO consistently achieved performance gains. In the table below, we provide a comparison of its performance against vanilla GRPO on these two models to further substantiate the validity of our concept and method.
>
> |                |              | Avg. | MathVerse | MathVision | MathVista | WE-MATH | DYNAMATH | GeoQA |
> | -------------- | ------------ | ---- | --------- | ---------- | --------- | ------- | -------- | ----- |
> | InternVL2.5-8B | Vanilla GRPO | 53.4 | 46.0      | 26.8       | 79.8      | 52.5    | 45.8     | 69.5  |
> |                | PS-GRPO      | 54.7 | 47.5      | 28.5       | 80.1      | 55.3    | 45.9     | 71.1  |
> | MultiMath      | Vanilla GRPO | 49.5 | 38.0      | 22.5       | 75.9      | 48.3    | 40.9     | 71.6  |
> |                | PS-GRPO      | 51.2 | 39.7      | 24.4       | 77.7      | 49.3    | 42.6     | 73.5  |
>
> > Q2: Confidence intervals
>
> Thank you for your question, which will help further demonstrate the robustness of the advantages in our experimental results. We have selected the strongest baselines from Table 1. The results are presented as follows:
>
> | **Model**               | **Avg**        | **MathVerse**  | **MathVision** | **MathVista**  | **WE-MATH**    | **DYNAMATH** | **GeoQA**      |
> | ----------------------- | -------------- | -------------- | -------------- | -------------- | -------------- | ------------ | -------------- |
> | Gemma3-12B              | 49.9 ± 0.2     | 40.1 ± 0.3     | 29.1 ± 0.2     | 63.6 ± 0.2     | 51.7 ± 0.3     | 45.8 ± 0.2   | 67.7 ± 0.2     |
> | Points-Qwen2.5-Instruct | 49.8 ± 0.2     | 41.1 ± 0.2     | 23.9 ± 0.2     | 76.2 ± 0.2     | 51.5 ± 0.6     | 42.9 ± 0.3   | 64.1 ± 0.2     |
> | AtomThink-EMOVA         | 49.5 ± 0.1     | 42.5 ± 0.2     | 24.9 ± 0.3     | 75.9 ± 0.5     | 49.6 ± 0.2     | 41.0 ± 0.2   | 70.5 ± 0.3     |
> | URSA-8B-PS-GRPO         | **58.1 ± 0.2** | **51.1 ± 0.4** | **31.7 ± 0.1** | **82.8 ± 0.5** | **61.2 ± 0.5** | **47.4 ± 0.2**   | **75.5 ± 0.5** |
>
> We will add in the revised version.
>
> > Q3: Provide confidence intervals
>
> Please see response to Q2.
>
> > Q4: Why employing different metrics
>
> Thanks for noticing this. The dataset descriptions in Table 1 are intended to clarify that we evaluate on different subsets of the same benchmark, not on different metrics. We followed the established practice for each dataset. For example, MathVision has a full test set of 3,040 examples and a smaller `testmini` split with 300. Prior work typically reports results on the full set, so we chose to align with that to ensure fairer comparison. We clarify this distinction in Table 1.
>
> > Q5: How sensitive is the "drop-moment" approach to hyperparameter selection?
>
> Please see our reponse to Q1.
>
> > Q6: Clarify that including BEL/MIE doesn't simply lead to performance improvement as more training samples are included
>
> Thanks for your question. In fact, before constructing DualMath, we attempted to validate the effectiveness of the PRM using PRM800K [1], and we report this result in the table below.
>
> |               | **MathVerse** |          |          |          | **MathVista** |          |          |          | **MathVision** |          |          |          |
> | ------------- | ------------- | -------- | -------- | -------- | ------------- | -------- | -------- | -------- | -------------- | -------- | -------- | -------- |
> |               | N=4           | N=8      | N=16     | N=32     | N=4           | N=8      | N=16     | N=32     | N=4            | N=8      | N=16     | N=32     |
> | PRM800K       | 49.5          | 50.1     | 50.8     | 51.0     | 82.3          | 82.7     | 83.0     | 83.2     | 29.9           | 30.6     | 31.1     | 31.4     |
> | Expanded BEL  | 53.1          | 53.5     | 53.9     | 54.2     | 82.2          | 84.7     | 85.2     | 85.9     | 30.9           | 32.7     | 33.6     | 34.1     |
> | DualMath-1.1M | **53.3**      | **54.2** | **54.7** | **55.0** | **83.2**      | **85.5** | **86.5** | **87.2** | **31.6**       | **33.1** | **34.0** | **35.1** |
>
> Furthermore, we considered randomly sampling a number of examples from BEL equal to that of MIE. We employed qwen3-32B to perform single-step inference rewriting, instructing the model to only modify the diversity of the textual expression without altering any content related to logical reasoning. This technique has also recently been utilized by Kimi-K2. [2] We believe this helps to validate that, given an equivalent data volume, our proposed contribution of focusing on image-text consistency in multimodal scenarios is reasonable.
>
> We appreciate your recognition of our work, and we hope the above content has addressed your concerns. If you have any further questions, please feel free to discuss.
>
> [1] Lightman, Hunter, et al. Let's Verify Step by Step.
>
> [2] Kimi-K2 Technical Report. Moonshot AI.

---

### Decision · Program_Chairs · 2025-09-17

**Decision:**

Accept (poster)

**Comment:**

The paper proposes a three-stage pipeline for multimodal math reasoning: a curated CoT corpus to warm up a base MLLM, a process-supervision set built with binary error locating and misinterpretation insertion, and PS-GRPO, which converts PRM drop-moment signals into an outcome penalty during online RL.
Reviewers are split. E38C and 4akc find the paper clear and empirically strong, yet question conceptual novelty. U6mC and GTaC raise concerns about cost, theoretical grounding, quality and bias of synthetic process labels, and presentation issues. The rebuttal seems to address several of these concerns: it adds confidence intervals and sensitivity studies, compares PRM to ORM on the same data, benchmarks PS-GRPO against step-level GRPO and direct process-reward variants, shows generalization across models, and provides LLM-as-judge checks plus ablations for the data construction.

Overall, soundness concerns are mostly addressed, and the added baselines strengthen the case. The paper is generally well executed, with broad experiments, released resources, and a practical RL tweak for multimodal PRMs; however, the contribution still may read as primarily integrative and a fairly straightforward extension of established techniques to the multimodal setting. The most distinctive idea is the specific reward shaping.